# Long-term cloud condensation nuclei number concentration, particle number size distribution, and chemical composition measurements at regionally representative observatories

Julia Schmale[1], Silvia Henning[2], Stefano Decesari[3], Bas Henzing[4], Helmi Keskinen[5,6], Karine Sellegri[7], Jurgita Ovadnevaite[8], Mira L. Pöhlker[9], Joel Brito[10,7], Aikaterini Bougiatioti[11], Adam Kristensson[12], Nikos Kalivitis[11], Iasonas Stavroulas[11], Samara Carbone[10], Anne Jefferson[13], Minsu Park[14], Patrick Schlag[14,16], Yoko Iwamoto[17,18], Pasi Aalto[5], Mikko Äijälä[5], Nicolas Bukowiecki[1], Mikael Ehn[5], Göran Frank[12], Roman Fröhlich[1], Arnoud Frumau[19], Erik Herrmann[1], Hartmut Herrmann[2], Rupert Holzinger[15],
Gerard Kos[19], Markku Kulmala[5], Nikolaos Mihalopoulos[11,20], Athanasios Nenes[21,20,22], Colin O'Dowd[8], Tuukka Petäjä[5], David Picard[7], Christopher Pöhlker[9], Ulrich Pöschl[9], Laurent Poulain[2], André Stephan Henry Prévôt[1], Erik Swietlicki[12], Meinrat O. Andreae[9], Paulo Artaxo[10], Alfred Wiedensohler[2], John Ogren[13], Atsushi Matsuki[17], Seong Soo Yum[14], Frank Stratmann[2], Urs Baltensperger[1], and Martin Gysel[1]

[1] Laboratory of Atmospheric Chemistry, Paul Scherrer Institute, 5232 Villigen, Switzerland
[2] Leibniz Institute for Tropospheric Research, Permoserstrasse 15, 04318 Leipzig, Germany
[3] Institute of Atmospheric Sciences and Climate, National Research Council of Italy, Via Piero Gobetti, 101, 40129 Bologna, Italy
[4] Netherlands Organisation for Applied Scientific Research, Princetonlaan 6, 3584 Utrecht, The Netherlands
[5] Department of Physics, University of Helsinki, Gustaf Hällströmin katu 2, 00014, Helsinki, Finland
[6] Hyytiälä Forestry Field Station, Hyytiäläntie 124, Korkeakoski, Finland
[7] Laboratory for Meteorological Physics (LaMP), Université Clermont Auvergne, F-63000 Clermont-Ferrand, France
[8] School of Physics and CCAPS, National University of Ireland Galway, University Road, Galway, Ireland
[9] Multiphase Chemistry and Biogeochemistry Departments, Max Planck Institute for Chemistry, Mainz, Germany
[10] Instituto de Física, Universidade de São Paulo, Rua do Matão 1371, CEP 05508-090, São Paulo, SP, Brazil
[11] Department of Chemistry, University of Crete, Voutes, 71003 Heraklion, Greece
[12] Department of Physics, Lund University, 221 00 Lund, Sweden
[13] Earth System Research Laboratory, National Oceanic and Atmospheric Administration, 325 Broadway, Boulder, CO
80305, USA
[14] Department of Atmospheric Science, Yonsei University, Seoul, South Korea
[15] Institute for Marine and Atmospheric Research, University of Utrecht, Utrecht, The Netherlands
[16] Institute for Energy and Climate Research (IEK-8): Troposphere, Forschungszentrum Jülich, Jülich, Germany
[17] Institute of Nature and Environmental Technology, Kanazawa University, Kakuma-machi, Kanazawa 920-1192, Japan
[18] Graduate School of Biosphere Science, Hiroshima University, 1-4-4, Kagamiyama, Higashi-Hiroshima 739-8528, Japan
[19] Energy Research Center of the Netherlands, Petten, The Netherlands
[20] National Observatory of Athens, P. Penteli 15236, Athens, Greece
[21] School of Chemical & Biomolecular Engineering and School of Atmospheric Sciences, Georgia Institute of Technology, Atlanta, GA, 30332-0340, USA
[22] Foundation for Research and Technology – Hellas, Greece

*Correspondence to*: Julia Schmale (julia.schmale@gmail.com) and Martin Gysel (martin.gysel@psi.ch)

**Abstract.**

Aerosol-cloud interactions (ACI) constitute the single largest uncertainty in anthropogenic radiative forcing. To reduce the uncertainties and gain more confidence in the simulation of ACI, models need to be evaluated against observations, in particular against measurements of cloud condensation nuclei (CCN). Here we present a data set - ready to be used for model validation - of long-term observations of CCN number concentrations, particle number size distributions and chemical composition from twelve sites on three continents. Studied environments include: coastal background, rural background, alpine sites, remote forests and an urban surrounding. Expectedly, CCN characteristics are highly variable across site categories. However, they also vary within them, most strongly in the coastal background group, where CCN number concentrations can vary by up to a factor of 30 within one season. In terms of particle activation behavior, most continental stations exhibit very similar activation ratios (relative to particles > 20 nm) across the range of 0.1 to 1.0 % supersaturation. At the coastal sites the transition from particles being CCN inactive to becoming CCN active occurs over a wider range of the supersaturation spectrum..

Several stations show strong seasonal cycles of CCN number concentrations and particle number size distributions, e.g., at Barrow (Arctic Haze in spring), at the alpine stations (stronger influence of polluted boundary layer air masses in summer), the rain forest (wet and dry season), or Finokalia (forest fire influence in fall). The rural background and urban sites exhibit relatively little variability throughout the year while short-term variability can be high especially at the urban site.

The average hygroscopicity parameter, $\kappa$, calculated from the chemical composition of submicron particles, was highest at the coastal site of Mace Head (0.6) and the lowest at the rain forest station ATTO (0.2 - 0.3). We performed closure studies based on the $\kappa$-Köhler theory to predict CCN number concentrations. The ratio of predicted to measured CCN concentrations is between 0.87 and 1.4 for five different types of $\kappa$. The temporal variability is also well captured, as reflected by Pearson correlation coefficients > 0.87.

Information on CCN number concentrations at many locations is important to better characterize ACI and their radiative forcing. But long-term comprehensive aerosol particle characterizations are labor intensive and costly. Hence, we recommend operating "migrating-CCNCs" to conduct collocated CCN number concentration and particle number size distribution measurements at individual locations throughout one year at least to derive a seasonally resolved hygroscopicity parameter. This way, CCN number concentrations can be calculated based on continued particle number size distribution information only and greater spatial coverage of long-term measurements can be achieved.

**1 Introduction**

Cloud droplets are formed by activation of a subset of aerosol particles called cloud condensation nuclei (CCN), which affect

the radiative properties of clouds through modifying the cloud droplet number concentrations (CDNC), cloud droplet size, cloud lifetime and precipitation processes (Rosenfeld et al., 2014). To date, radiative forcing through aerosol-cloud interactions constitutes the least understood anthropogenic influence on climate (IPCC, 2013): the uncertainty in aerosol-induced radiative forcing of $\pm$ 0.70 (W m$^{-2}$) (from a mean of -0.55 W m$^{-2}$) is twice the uncertainty for $CO_2$ ($\pm$ 0.35, mean +1.68 (W m$^{-2}$)). This uncertainty propagates through to e.g., the calculation of climate sensitivity, a variable that is needed to

predict global temperature increase for given emission scenarios (Andreae et al., 2005; Seinfeld et al., 2016). It remains a significant challenge to reduce these uncertainties and to increase thereby our confidence in global and regional climate scenarios (IPCC, 2013; Lee et al., 2013; Seinfeld et al., 2016).

The number concentration of CCN is not the only factor determining the CDNC. Reutter et al. (2009) found that cloud droplet formation can either be limited by the presence of CCN (CCN-limited regime), by the updraft velocity (updraft-

limited regime), or both (transition regime). Globally, however, the CCN-limited regime prevails (Rosenfeld et al., 2014). Among the main factors driving the uncertainty in simulating CCN abundance are the aerosol particle number size distributions, size-dependent removal processes, the contribution of boundary layer new particle formation events to particle number concentration and their size, the particle number size distribution of emitted primary particles, the particle activation diameter, the formation of biogenic and anthropogenic secondary organic aerosol (SOA) as well as the processing of $SO_2$ in

clouds into particulate sulfate (e.g., Croft et al., 2009; Lee et al., 2013; Wilcox et al., 2015). Information on aerosol hygroscopicity is also needed to constrain uncertainty (Rosenfeld et al., 2014). These factors affect the ability of aerosol particles to form CCN at both large-scale and long-term periods as well as at the regional scale and in the short-term.

To improve model performance, data from measurements of particle number size distribution, CCN number concentrations, aerosol particle chemical composition and hygroscopicity are needed (Carslaw et al., 2013; Ghan and Schwartz, 2007;

Rosenfeld et al., 2014; Seinfeld et al., 2016). Satellite observations, covering large scales and longtime horizons, can provide proxies of these variables. However, the resolution is often too coarse to study detailed aerosol-cloud interactions (Rosenfeld et al., 2014; Rosenfeld et al., 2016; Shinozuka et al., 2015) and other shortcomings exist. For example, a common proxy is aerosol optical depth (AOD). It has been found that the correlation of AOD with CCN number concentrations, a key assumption in this approach, is strongly dependent on ambient relative humidity and aerosol types. Furthermore, these

correlations become less reliable when sea salt and mineral dust constitute an important fraction of the particle number, a situation which can be relevant over the ocean or deserts (Liu and Li, 2014). This makes in-situ measurements indispensable and therefore numerous studies of CCN activity have been carried out in a variety of environments, ranging from remote

marine over continental background to urban locations, and in the laboratory (e.g., Andreae, 2009a; Asmi et al., 2012; Bougiatioti et al., 2009; Crosbie et al., 2015; Cubison et al., 2008; Ervens et al., 2010; Jurányi et al., 2010; Paramonov et al., 2015; Rose et al., 2011; Whitehead et al., 2014; Wong et al., 2011). Most of these observations focus on relatively short time periods and some attempt to capture specific circumstances such as biomass burning events (e.g., Bougiatioti et al., 2016) or

focus on the hygroscopicity of specific aerosol particle components such as black carbon (e.g., Schwarz et al., 2015) or organic carbon (e.g., Frosch et al., 2011). While such studies provide detailed insights into CCN activation processes and contribute to our comprehensive understanding of aerosol-cloud interactions, they cannot address questions of regional and temporal CCN variability. However, those aspects are crucial for model evaluation. Also, knowledge of size distribution, composition and hygroscopicity of aerosol components and atmospheric aerosols in different environments as well as

appropriate representation in model simulations is important to quantify aerosol radiation interactions as a function of relative humidity .

They are best addressed through long-term observations at regionally representative locations. Among the scarce examples of such studies are observations at the high alpine site Jungfraujoch (Jurányi et al., 2011), in the Amazon rain forest (Pöhlker et al., 2016), or several other European stations (Mace Head, Ireland, coastal background; Hyytiälä, Finland, boreal forest;

Vavihill, Sweden, rural background) before or during the European Integrated project on Aerosol Cloud Climate and Air Quality Interactions (EUCAARI) experiment (Fors et al., 2011; Paramonov et al., 2015; Sihto et al., 2011). Further examples of long-term studies include a study at an the urban background site in Vienna, Austria (Burkart et al., 2011), at a regionally representative site in the Yangtze River Delta (Che et al., 2016), or at an urban site in Shanghai (Leng et al., 2013). In addition to revealing the seasonal and regional variability of CCN number concentrations and associated variables, such

long-term studies can address the question of which specific aerosol particle characteristics need to be monitored to provide data sets with which models can be effectively evaluated. Such studies are particularly valuable given general constraints that will not allow operating very comprehensive aerosol characterization equipment over long periods of time at many locations. One specific question is whether CCN number concentrations need to be measured directly, e.g., with cloud condensation nuclei counters (CCNC) or whether they can be inferred by knowing the critical diameter at which particles

activate as cloud droplets. A simple parameterization was developed from the $\kappa$-Köhler theory (Petters and Kreidenweis, 2007), which links aerosol particle hygroscopicity with the critical diameter at a given supersaturation and hence leaves the particle number size distribution as determining variable for CCN number concentrations. The hygroscopicity parameter, $\kappa$, can be calculated from the aerosol particle chemical composition. So theoretically, it would not be necessary to operate a CCNC if particle number size distribution and chemical composition measurements were available. This, however, leads to

the question of which degree of detail is needed for the chemical composition and mixing-state of the aerosol particles to derive their hygroscopicity. However, there is no unanimous conclusion in the literature. Some studies find that the variability in aerosol size distribution is more important than the variability in chemical composition (e.g., Dusek et al., 2006; Ervens et al., 2007) and a review (Andreae and Rosenfeld, 2008) suggests that a global hygroscopicity parameter of $\kappa$ = 0.3 ± 0.1 and $\kappa$ = 0.7 ± 0.2 can be useful as a first approximation for continental and marine aerosol, respectively.

Conversely, other studies stress the importance of not only knowing the bulk composition of particles, but also their size-resolved chemical composition and state of mixing or even the more detailed composition of organic carbon. This is because organic aerosol usually constitutes an important fraction of the CCN relevant aerosol mass around the globe (Zhang et al., 2007) and more oxygenated aerosol tends to be more hygroscopic (Cubison et al., 2008; Duplissy et al., 2008; Frosch et al.,

2011; Jimenez et al., 2009; Massoli et al., 2010; Wong et al., 2011). In addition, several studies have investigated the effect of organic surfactants that can decrease the surface tension (e.g., Charlson et al., 2001; Facchini et al., 2000). It is expected that the effect of surface tension suppression by surfactants is smaller than predicted by the classical Köhler-theory due to surface-bulk partitioning effects unless liquid-liquid phase separation occurs (Sorjamaa et al., 2004). A recent study shows that a combination of liquid-liquid phase separation, surfactants and specific particle size distributions could, however,

increase the CCN number concentration by a factor of ten compared to climate model predictions (Ovadnevaite et al., 2017). More generally, the importance of a detailed knowledge of the particle chemical composition for CCN activity depends on the distance from the source as more aged particles tend to assume similar particle number size distributions and hygroscopic characteristics (e.g., Andreae, 2009b; Ervens et al., 2010).

In this study, we present long-term observations from twelve locations of collocated particle number size distributions, CCN number concentrations, and in some cases aerosol particle chemical composition measurements. Eight of these stations are part of the European Aerosols, Clouds, and Trace gases Research InfraStructure (ACTRIS, http://www.actris.eu/), while the other observatories are located in Korea, Japan, the USA, and Brazil. They cover a range of environments such as coastal and rural background, urban and high alpine conditions, as well as boreal, Arctic and rain forest characteristics. We explore

the frequency distributions and seasonal cycles of various variables (CCN number concentration, critical diameter, $\kappa$ values, and others), the persistence of CCN number concentrations in winter and summer, and particle activation behavior. We also perform closure studies based on the $\kappa$-Köhler parameterization and test the sensitivity of results to simplified assumptions regarding aerosol chemical composition, particle number concentrations and size distributions.

## 2 Methodology

### 2.1 Measurement sites and instrumentation

Figure 1a shows the locations of the twelve observatories, which span a wide range of environments. Four stations are located near the coast, covering Arctic (BRW), Mediterranean (FIK), Atlantic (MHD), and Pacific conditions (NOT). Two

alpine stations in Europe (PUY, JFJ) represent the continental background and partly free tropospheric air masses, while three observatories near sea level in Europe characterize the rural background conditions (MEL, CES, VAV). The boreal

(SMR) and rain forest (ATT) environments are represented by one station each, as well as one urban location in Asia (SEO) (compare grouping in Fig. 1b). Table 1 provides an overview of each station's characteristics and representativeness.

This study uses data from concomitant measurements of CCN number concentrations, particle number size distributions, and, where available, bulk, aerosol particle chemical composition. Table 1 lists instrumental and operational details. All information regarding each station's inlet system, instrument descriptions and sampling details is given in the related data descriptor paper (Schmale et al., 2017), except for the rainforest station (ATTO), which is described in Pöhlker et al. (2016). Since the focus is on long-term observations rather than short-term intensive field campaigns, the data used were chosen to cover at least 75 % of each season within twelve consecutive months. Seasons are defined as December, January, February (DJF), March, April, May (MAM), June, July, August (JJA), and September, October, November (SON) if not referred to otherwise.

Briefly, CCN number concentrations were measured with the CCNC-100 model by Droplet Measurement Technologies in all cases except at Puy de Dôme, where a miniature version of this instrument was deployed (Sullivan et al., 2009). Most stations measured in the polydisperse mode, where the activation of the entire aerosol population is measured at a given supersaturation. At four stations (ATT, MEL, PUY, NOT), CCN number concentrations were determined in the monodisperse mode, whereby particles are selected by a differential mobility analyzer (DMA) that scans through a range of particle diameters upstream of the CCNC. Regardless of the operation mode, this work considers exclusively the time series of the bulk activated aerosol, meaning that monodisperse CCN number concentrations were integrated over the covered size ranges.

Particle number size distributions were obtained by a variety of mobility particle size spectrometers (MPSS) as listed in Table 1, which are either commercially available or custom-built. All custom-built versions have been intercompared at the World Calibration Center for Aerosol Physics (Wiedensohler et al., 2012) or audited by it.

Submicron aerosol particle chemical compositions were measured by two different types of aerosol mass spectrometers. The high-resolution time-of-flight aerosol mass spectrometer (HR-ToF-AMS) operated at Mace Head has been described by DeCarlo et al. (2006) in general and in particular for Mace Head by Ovadnevaite et al. (2014). The aerosol chemical speciation monitor (ACSM), deployed at all other stations, has been introduced by Ng et al. (2011) and the first official ACTRIS intercomparison is described in Crenn et al. (2015). The intercomparison covers all quadrupole ACSMs, except the one deployed at ATTO, which is described in Pöhlker et al. (2016). On Jungfraujoch, a time-of-flight ACSM was operated as described by Fröhlich et al. (2015; 2013). All aerosol mass spectrometer types are able to provide the mass concentrations of standard chemical species that include particulate ammonium, chloride, nitrate, organics, and sulfate in the submicron size range. Table 1 lists which species are available at each station; missing species mean that their concentrations were below detection limit. At Mace Head, the sea salt content of the submicrometer aerosol is given in addition based on a specific method introduced by Ovadnevaite et al. (2012). Table 1 also lists the collection efficiency (CE) of each mass spectrometer. The CE depends on the transmission of particles into the instrument and their chemical composition and is hence an instrument and site specific factor (Huffman et al., 2005; Middlebrook et al., 2012).

Additionally, at the time of data collection, equivalent black carbon (BC) mass concentrations were available for the stations JFJ (aethalometer model AE31, Magee Scientific), MEL and MHD (multi-angle absorption photometer, MAAP, Thermo Scientific), which are used for the sole purpose of calculating the hygroscopicity parameter $\kappa$ (see section 2.3.2). For stations where no concomitant BC concentration time series were available, BC mass fractions from the literature were used as approximation as described in section 2.3.2.

## 2.2 Data treatment and quality assurance

The collection, harmonization and quality assurance of the data sets presented here are described in detail in the data descriptor by Schmale et al. (2017). Data have a time resolution of one hour and represent standard temperature and pressure (STP) conditions. The time resolution of CCN number concentrations at Puy de Dôme (PUY) and ATTO are four and six hours, respectively, because the scans over the submicron aerosol size range in monodisperse mode took longer. Most instruments measuring particle number size distributions had either been intercompared, audited, or the data had been published previously, see Table 9 in Schmale et al. (2017). The same was the case for the chemical composition data (same reference). For that reason, emphasis was given to the quality check of the CCN number concentrations that had not previously been published in most cases. Exceptions are the data from Seoul (Kim et al., 2014) and ATTO (Pöhlker et al., 2016), whereby the latter station is not included in Schmale et at. (2017). Note that the aerosol sample flow was kept at a relative humidity < 40 % at all sites except in Seoul, meaning that particle size can be biased large. For all polydisperse data sets where measurements at a supersaturation of 1.0 % were available, the total CCN number concentration was compared to the total particle number concentration in all instances when the contribution of particles < 30 nm was at most 10 %. It is expected that at such a high supersaturation almost all particles > 30 nm activate. Hence the data points are expected to group around the 1:1 line within the target uncertainty of 10 % (Wiedensohler et al., 2012). Figure 4 in Schmale et al. (2017) shows that most instruments performed reasonably well, with the exception of the CCNCs at the Cesar (CES) and Jungfraujoch (JFJ) stations. At CES the CCN number concentration is strongly underestimated, and the underestimation increases with increasing supersaturation. Discrepancies are as large as a factor of 3.3 in the geometric mean for 1.0 % supersaturation. This suggests that small particles, activating at higher supersaturation, were not sufficiently accounted for by the CCNC. As this was not due to insufficient droplet growth to the detection limit of 1 µm of the optical particle counter in the CCNC, the bias most likely originated from particle losses in the sampling line to the CCNC. Since this cannot be accounted for across the various supersaturations, the dataset has not been corrected. Therefore the CCN number concentrations reported for CES represent a lower limit. Details for JFJ have already been discussed in (Schmale et al., 2017). More details for both stations are provided in the supplementary information section 1 (hereafter referred to as SI Sec 1).

At the observatories in Melpitz (MEL), NOT and PUY, CCN were not measured at a supersaturation of 1.0% but in monodisperse mode. Therefore, the integrated particle number concentration above the critical diameter at a measured supersaturation (diameter at which particles activate) was plotted against the integrated CCN number above the same

diameter. The CCN number size distribution data at both stations compare well with the particle number size distributions (see Figure 5 in Schmale et al., 2017).

All data (except for ATTO) are available from: http://actris.nilu.no/Content/products. The ATTO data have been published
by Pöhlker et al. (2016).

### 2.3 Data analyses

### 2.3.1 Frequency distributions, seasonal cycles and persistence

The CCN number concentration frequency distributions were calculated in 200 bins with a logarithmic ($\log_{10}$) spacing of
0.023, starting with 1 particle (cm$^{-3}$). Frequency distributions of the particle number size distributions' geometric mean diameter ($D_g$) were calculated for the available particle diameter range at each station, and also starting at a lower cut-off of 20 nm for comparability. The frequency distributions of $D_g$ as well as the critical diameters ($D_{crit}$) are based on 105 bins with a logarithmic ($\log_{10}$) spacing of 1/64. The value of $D_{crit}$ was derived from integrating the particle number size distributions from their maximum diameters to that diameter at which the integrated particle number equaled the measured CCN number
concentration (see also section 2.3.2, Eq. 5). All frequency distributions are normalized to the number of data points at each station.

Seasonal cycles are represented by the monthly medians calculated from the hourly values of the respective variable (four and six hourly data for CCN at PUY and ATT, respectively). If a particular month is covered several times in a time series, the median of all data acquired in that month is derived. Additionally, the interquartile range has been calculated in the same
way.

The CCN number concentrations at the regionally representative stations discussed here are influenced by a variety of factors that include: the microphysical and chemical characteristics of the particles, atmospheric transport, dry and wet particle deposition, synoptic patterns as well as seasonal source strengths. For example, the boreal forest produces more secondary organic aerosol (SOA) in the growing season (summer) than in winter. Determining the persistence of CCN
number concentrations, i.e., the duration over which their concentration remains similar, can help to identify regionally relevant factors that significantly influence the abundance of CCN. At each station, the persistence was calculated by auto-correlating the time series for the winter (DJF) and summer (JJA) months. Data gaps of less than one day were filled by the average of the preceding four data points. Large data gaps, exceeding one day, were not filled. Instead shorter periods of the season were auto-correlated separately and then averaged. This was the case for JFJ and BRW in winter, and MHD, FIK, and
BRW in summer. The auto-correlation function *acf* in the program R (version 3.3.1) was applied to the time series of CCN at a supersaturation of 0.2 % with one hour time resolution, except for ATT and PUY where the highest time resolutions were six and four hours, respectively. The significance level of the auto-correlation was determined by calculating the large lag

standard error, $E_{corr}$, of the auto-correlation coefficient, accounting for the interdependency between auto-correlation coefficients, following Eq. (1):

$$E_{corr}(r_k) = \sqrt{\frac{1}{N}\left(1 + 2\sum_{i=1}^{K} r_i^2\right)} \tag{1}$$

with $N$ being the number of data points, $r_k$ the correlation coefficient at lag $k$, and $K < k$, with $K$ being the last lag of a specific calculation step. The higher the number of observations, the larger $E_{corr}$ becomes and with this, the likelihood of identifying a potentially randomly high correlation at a large lag as significant. The persistence of a property is determined by the time coordinate at which the auto-correlation curve crosses the large lag standard error curve.

### 2.3.2 Hygroscopicity parameter kappa ($\kappa$) and CCN closure

The hygroscopicity parameter, $\kappa$, quantifies the Raoult effect, i.e., the relationship between the particle's hygroscopic equilibrium growth factor ($GF$) and corresponding water activity. When assuming a surface tension and using the Köhler equation, which combines the Raoult and Kelvin effects to the related growth factor and RH at equilibrium, the $\kappa$ value unambiguously relates the dry particle size with the critical supersaturation (Petters and Kreidenweis, 2007): the higher the value of $\kappa$, the higher the hygroscopicity of a particle (Zieger et al., 2017). The $\kappa$ of a mixed particle can be derived in good approximation from the particle chemical composition following a simple mixing rule as given in Eq. (2) when the $\kappa$ value of each component $i$ is known (Petters and Kreidenweis, 2007):

$$\kappa = \sum_i \varepsilon_i \kappa_i \tag{2}$$

with $\varepsilon_i$ being the volume fraction of component $i$. The volume fraction of each component was derived from its measured mass concentrations and density (1.4 g cm$^{-3}$ was assumed for organic aerosol) in this work.

The $\kappa$ values of pure substances typically depend on water activity. Petters and Kreidenweis (2007) provide $\kappa$ values for a variety of chemical components including inorganic salts and acids. These, however, only partly refer to conditions at the point of particle activation. We therefore calculated the pure component $\kappa$ values for a reference water activity of $a_w$=0.9975 following Petters and Kreidenweis (2007):

$$\frac{1}{a_w} = 1 + \kappa \frac{V_s}{V_w} \tag{3}$$

$V_s$ is the volume of the dry particulate matter and $V_w$ the volume of water. The reference $a_w$ was chosen to reflect the water activity in the solution droplet at the point of CCN activation for a supersaturation of 0.5 %, temperature of 5 °C and corresponding pure water surface tension of 74.95 (mN m$^{-1}$) and $\kappa$ of 0.3. These properties and conditions are typical for cloud formation in ambient clouds and they imply a critical dry particle diameter of 63 nm. Note that the temperature has only a minimal effect on the $\kappa$ of a pure component, while it affects CCN activation through the temperature dependence of surface tension and the Kelvin effect. This reference water activity was used as input to the E-AIM model II and IV (http://www.aim.env.uea.ac.uk/aim/aim.php), by which the particulate water content was calculated for the pure salts and

inorganic acids in aqueous solution. The E-AIM II model is an equilibrium thermodynamic model including the following ions: $H^+$, $NH_4^+$, $SO_4^{2-}$, $NO_3^-$, $H_2O$. It is valid from 328 K to about 200 K. Model IV includes $Na^+$ and $Cl^-$ and is valid from 180 - 330 K. Based on this, the *GFs* and from that the $\kappa$ values were calculated for sulfuric acid, ammonium sulfate, ammonium bisulfate, ammonium nitrate and sodium chloride, accounting for the solution density which is provided by the

AIM model. Note, we did not account for the water content of the chemical species in dry conditions, e.g., RH = 10 %. The chemical species were derived from ions quantified by the mass spectrometric measurements following the procedure suggested by Gysel et al. (2007). The results (shown in Fig. 2) are generally similar to and slightly lower than the ideal $\kappa$ ($a_w$= 1), but can be larger or smaller than the values provided in Petters and Kreidenweis (2007). Note that the value for NaCl in Petters and Kreidenweis is too low, instead of 1.12 it should be around 1.5 (Zieger et al., 2017).

In our study, we assume that chloride is present in the form of NaCl and apply the $\kappa$ value as shown in Fig. 2. For MHD, the contribution of submicron sea salt has been calculated by the data originators after Ovadnevaite et al. (2011a) to which we assign the same $\kappa$ value. Given that the AMS and ACSM do not fully detect sea salt components which are present in the submicron aerosol (Salter et al., 2015), this contribution is sea salt mass contributions are likely to be underestimated at all other stations close to the sea and where chemical composition data are available (e.g., CES, FIK). ) except at MHD.

For particulate organics, we use a $\kappa$ of 0.1, following observations in a variety of environments (e.g. Dusek et al., 2010; Gunthe et al., 2009; Gunthe et al., 2011; Juranyi et al., 2009; Rose et al., 2011; Rose et al., 2010). It should be noted, however, that $\kappa_{org}$ has been observed to be higher in other studies, especially when the organic aerosol becomes more oxygenated that is when chemical aging has taken place (e.g., Chang et al., 2010; Massoli et al., 2010). At an O:C ratio of 0.2, i.e. non-oxygenated organic matter, $\kappa_{org}$ tends to be < 0.10, while it increases towards 0.25 or higher at a ratio near 1.0

(e.g., Wong et al., 2011). At some forest sites, significant organic particle mass is produced in-situ and the atmospheric processing during transport might have only a small influence. A previous study in the Amazon rain forest revealed that the $\kappa$ value of the bulk aerosol is only slightly larger than 0.1, when the organic aerosol mass fraction is close to 1. At the boreal forest site (SMR), however, the $\kappa$ value seems to fall in between 0.1 and 0.2 for high organic mass fractions (Paramonov et al., 2013). It is conceivable that in-situ contribution to organic aerosol mass from biogenic emissions are smaller than in the

rain forest, and that hence forest emissions upwind that are transported and chemically processes over hours to days play a larger role in determining $\kappa_{org}$. At sites like CES, which are classified as background sites but located relatively close to urban agglomerations (20 and 30 km from Rotterdam and Utrecht, respectively), the observed organic matter might have been processed sufficiently to become more hygroscopic than what is normally observed in the urban environment (e.g., Ervens et al., 2010). For black carbon (BC) we use $\kappa = 0$ (e.g., Hitzenberger et al., 2003; Rose et al., 2011; Tritscher et al.,

30   2011).

With these $\kappa$ values for individual components, we calculate the bulk aerosol hygroscopicity with Eq. 2 in five variations:

1.   deriving all chemical components, including salts and acids, using the ammonium, nitrate, sulfate and chloride ions, and organics from the aerosol chemical composition data, and no BC (referred to as '$\kappa_{IA+OA-BC}$');

2. only with ion-balanced (IB) inorganic components, which excludes acids and bisulfates, but with organics, and no BC ('$\kappa_{IB+OA-BC}$');

3. similar to 1 but including BC ('$\kappa_{IA+OA+BC}$');

4. similar to 2 but including BC ('$\kappa_{IB+OA+BC}$');

5. $\kappa = 0.3$.

For alternatives 2 and 4, the measured number of sulfate and nitrate ions was neutralized with a calculated amount of ammonium. We chose to calculate ammonium because the quantification of ammonium with the aerosol mass spectrometer is subject to higher uncertainty than for sulfate and nitrate. Chloride was assumed to be present as sodium chloride. All particulate sulfate and nitrate were assumed to be inorganic, because most composition data were obtained from unit mass resolution ACSM measurements, which do not allow apportioning these ions to organic species. The contribution of particulate sulfate to ammonium sulfate, ammonium bisulfate, and sulfuric acid were obtained after equation 2 in Gysel et al. (2007) when using prediction alternative 1 and 3. For the stations MEL, MHD and JFJ, BC time series were available. For stations where no BC time series were available at the point of data collection, seasonal or yearly average values were taken from the literature. For ATTO, BC concentrations were obtained from Fig. 30 in Andreae et al. (2015), for CES from Schlag et al. (2016), for SMR from Hyvärinen et al. (2011), and for FIK from Bougiatioti et al. (2014). Results for all $\kappa$ values are provided in Table 2. It must be noted that when using bulk aerosol particle chemical composition data from AMS or ACSM measurements, the larger particles (all instruments used inlet lenses with an upper cut-off of 1 µm) will dominate the aerosol mass. Hence, the composition information is representative of the size range around the peak of the mass size distribution and might not reflect the composition of the majority of particles when small particles dominate the number concentration. This can be the case when new particle formation happens, e.g., at SMR or MEL (Manninen et al., 2010). In the presence of mostly accumulation mode particles, however, good agreement between hygroscopic growth factor measurements and its derivation from bulk aerosol composition data has been found for SMR, e.g., Raatikainen et al. (2010). At JFJ earlier studies deriving $\kappa$ from hygroscopic tandem DMAs and the CCNC resulted in $\kappa = 0.20$ and $0.24$ (Jurányi et al., 2011; Kammermann et al., 2010a, respectively), showing that the method of deriving the particle hygroscopicity can play a role at some locations.

The size of the particles is an even more important determining factor for a particle's ability to act as CCN than the $\kappa$ value. Hence, for all stations where particle number size distribution and chemical composition data are available, we can predict the number of CCN particles at a given SS using the $\kappa$-Köhler equation (Eq. 6, Petters and Kreidenweis, 2007). This equation describes the equilibrium saturation ratio $S$ (ratio of the partial vapor pressure of water and the saturation vapor pressure of water) over an aqueous solution droplet:

$$S = \left(1 + \kappa \frac{D_0^3}{D_{drop}^3 - D_0^3}\right)^{-1} exp\left(\frac{4\sigma_{sol}\vartheta_w}{RTD_{drop}}\right) \tag{4}$$

with $D_0$ being the dry particle diameter, $D_{drop}$ the droplet diameter, $\sigma_{sol}$ the surface tension of the solution (we use a surface tension of water of 72.86 (mN m$^{-1}$) corresponding to 20 °C, which is close to the sample air temperature in the CCNC), $\vartheta_w$

the partial molar volume of water in the solution (which was assumed to be the molar volume of pure water), $R$ the universal gas constant, and $T$ the temperature. The first term on the right hand side of the equation is a semi-empirical formulation of the Raoult term, i.e. for the water activity $a_w$ expressed with dry size, droplet size and $\kappa$ value. More details are given elsewhere (e.g. Jurányi et al., 2010; Petters and Kreidenweis, 2007). The maximum of Eq. 4, with $D_{drop}$ being the independent variable, describes the critical supersaturation for a particle with given dry size and $\kappa$ value. Similarly, the critical dry diameter ($D_{crit}$) for a certain supersaturation and $\kappa$ value describes the dry size for which the corresponding critical supersaturation equals this supersaturation. The critical dry diameter was numerically derived from Eq. 4 (rather than using simplified and approximate analytical solutions).

Having determined $D_{crit}$ at a given SS and with assuming equal composition of all particles with similar size, we can calculate the number of activated particles by integrating the particle number size distribution from its maximum diameter ($D_{max}$) down to $D_{crit}$ following Eq. 5:

$$N_{CCN}(SS) = -\int_{D_{max}}^{D_{crit}(SS)} \frac{dN(D)}{dlogD} dlogD \qquad (5)$$

$N_{CCN}$(SS) can then be compared to the number of CCN at the same SS measured by the CCNC (i.e., a closure study).

At stations with simultaneous particle number size distribution and polydisperse CCN measurements, $\kappa$ can alternatively be derived by first estimating $D_{crit}$ with Eq. 5. This approach is only approximate for externally mixed aerosols. However, assuming a sharp activation cut-off, which is a priori incorrect in such cases, results in largely compensating errors (Kammermann et al., 2010a), thus still providing valid results.

## 3 Results and Discussion

### 3.1 Frequency distributions, seasonal cycles and persistence

Figure 1b provides an overview of CCN number concentration at SS = 0.2 % (CCN$_{0.2}$) per season at each station. Colored bars indicate the median while the black bars are a surrogate for seasonal variability spanning the interquartile range. The observatories are grouped by their stations classification (see colored shadings). It becomes apparent that there can be a large variability of CCN$_{0.2}$ number concentrations within one station category. Within the coastal background station category, the median values can be < 100 cm$^{-3}$ at BRW and higher than 1500 cm$^{-3}$ at NOT in spring. In the rural background category, in spring the largest difference is found between MEL with about 1600 (cm$^{-3}$) and VAV with about 400 (cm$^{-3}$). . Reasons are discussed in detail further below.

Fig. 3 shows normalized frequency distributions of CCN$_{0.2}$, the $D_g$ of the entire particle number size distributions (limited to sizes > 20 nm), and $D_{crit}$ at SS = 0.2 % as derived from Eq. 5 based on the CCN and particle number size distribution measurements only. The highest frequency of low CCN$_{0.2}$ number concentrations (< 200 cm$^{-3}$) can be found at the Arctic site

BRW, which is characteristic of the Arctic maritime environment (Barrie, 1986). Similarly low number concentrations are observed at the mountain sites PUY and JFJ with almost no contribution of > 1000 (cm$^{-3}$). This is expected as they represent continental background conditions as well as the free troposphere, mostly during winter and night-time, but also occasionally during summer (e.g., Herrmann et al., 2015; Venzac et al., 2009). Higher concentrations can be due to boundary layer air

mass injections, especially during summer. Note that potential influence from touristic activities was removed from the data sets (e.g. Fröhlich et al., 2015; Venzac et al., 2009). Low number concentrations are also found at the coastal site MHD (with the highest occurrence of CCN$_{0.2}$ densities of 200 cm$^{-3}$), which for certain periods reflects the clean marine conditions over the Atlantic Ocean (Ovadnevaite et al., 2014). The coastal environments of FIK in the Mediterranean and NOT in the Pacific Ocean exhibit generally higher concentrations (between 200 and 2000 cm$^{-3}$) due to particular pollution influences which for

example include long-range transport of NE European pollution and biomass burning plumes (Bougiatioti et al., 2016) and long-range transport of East Asian pollutions plumes (Iwamoto et al., 2016), respectively. In terms of CCN number concentrations, the NOT site is in fact similar to the European rural background sites MEL and CES, which experience higher concentrations than the higher latitude continental background site in VAV and the substantially cleaner boreal forest environment (SMR, both < 1000 cm$^{-3}$). The highest concentrations are seen in the urban environment of Seoul (SEO, 1000 –

5000 cm$^{-3}$). While CCN$_{0.2}$ concentrations are generally mono-modally distributed at all sites, the tropical rain forest observatory (ATT) and the Arctic station (BRW) exhibit bimodal distributions spanning a wide range of possible CCN number concentrations between 20 – 2000 cm$^{-3}$ and 20 – 200 cm$^{-3}$, respectively. As seen more clearly in the seasonal cycle (see Fig. 4), for ATTO this is due to the conditions of the rainy and dry seasons, as well as forest fires and other long-range transported air pollution influences (Pöhlker et al., 2016; Whitehead et al., 2016). At BRW the Arctic Haze period leads to

higher CCN number concentrations than observed in the remainder of the year.

Using $D_g$ as a proxy for aerosol size distributions, Fig. 3b shows that similar particle number size distributions do not always imply similar frequencies of CCN number concentrations. For example, the two mountain stations (JFJ, PUY) do not show similar frequency distributions of CCN$_{0.2}$ while they do for $D_g$, because the particle number concentration at PUY is higher and therefore more particles activate. BRW and MHD, while similar in their CCN$_{0.2}$ frequency distribution, exhibit

significantly different particle geometric mean diameters: mostly > 100 nm at BRW and mostly < 100 nm at MHD. The Nordic country stations (SMR, VAV) present similar particle number size distributions. This is true for the particle number size distributions with and without particles < 20 nm considered. The difference in results of $D_g$ when excluding particles < 20 nm is due to frequent new particle formation events at these locations (Manninen et al., 2010). The largest particles are observed in the most remote places, the Arctic (BRW) and the rain forest station (ATT) with high frequencies of $D_g > 100$

nm.

The critical diameters at SS = 0.2 %, being an indication for the particle hygroscopicity, as shown in Fig. 3c, provide yet another perspective on the diverse aerosol populations. Differences in aerosol sources might not necessarily be visible in the size distributions, whereas they can show up in terms of hygroscopicity. At a constant SS, a smaller $D_{crit}$ is expected for more hygroscopic particles such as sea salt. This is reflected by the $D_{crit}$ distributions at MHD and BRW. The distributions are

bimodal with high $D_{crit}$ occurrences of greater and smaller 100 nm, suggesting that the smaller mode is associated with sea salt and other CCN active marine aerosols in the case of MHD (Ovadnevaite et al., 2011b) and the generally highly hygroscopic Arctic background aerosol in BRW (Lathem et al., 2013). The second, less hygroscopic mode can be associated with a variety of other aerosol sources such as particles transported from inland sources which include peat combustion,

traffic and industrial emission sources (Ovadnevaite et al., 2011b; Taylor et al., 2016) for MHD, or industrial or biomass burning pollution plumes in the Arctic (Lathem et al., 2013). In the Mediterranean environment the distribution is not bimodal, although it exhibits a small plateau for slightly more hygroscopic particles around 100 nm, while the majority of particles are on average less hygroscopic (high $D_{crit}$ occurrence at 180 nm) than in the other coastal areas. This might be due to European pollution outflow and biomass burning plumes (Bougiatioti et al., 2016). At NOT, despite the influence of two

distinct sources, marine aerosol and long-range transported anthropogenic pollution (Iwamoto et al., 2016), only a mono-modal distribution of $D_{crit}$ is found (peak at 90 nm). This is likely due to the dominant wind direction from the west. Particles from different sources are hence continuously mixed and low-volatility gaseous components condense on all types of particles, which results in a mono-modal size distribution. This is different from MHD and BRW where different wind directions advect aerosol from different sources. At most other locations, the distributions of $D_{crit}$ are relatively narrow and

centered around or are slightly larger than 100 nm for SS = 0.2 %, except for JFJ. Here, also a second mode around 150 nm is found, most likely originating from boundary layer air mass injections in summer, as the seasonal cycle of $D_{crit}$ suggests in Fig. 4c. Investigation of diurnal cycles clearly shows that aerosol hygroscopicity decreases with boundary layer air mass injections due to changes in aerosol chemical composition (Jurányi et al., 2011; Kammermann et al., 2010a). Note that the second mode is likely over-weighted in Fig. 3c because there are more summers than winter seasons in the data set. In Fig. 4

monthly data were averaged and are hence equally weighted.

The seasonal cycles of $CCN_{0.2}$ number concentration, $D_g$, and $D_{crit}$ show characteristic differences between the locations (Fig. 4). As mentioned above, boundary layer air masses are uplifted in summer at JFJ, which is evident from the enhanced CCN number concentration, a median of 240 $cm^{-3}$ compared to about 20 $cm^{-3}$ in winter (compare also with Jurányi et al.,

2011) and the total particle number concentration (see SI Sec 3 for all stations). At the same time, particles are larger ($D_g$ about 75 nm in summer versus 50 nm in winter, panel (b)), but less hygroscopic ($D_{crit} > 100$ nm versus < 100 nm panel (c)). A similar seasonal cycle exists at PUY, however less pronounced, likely due to its lower elevation. Both forest environments also show seasonal cycles. In the boreal forest (SMR), $CCN_{0.2}$ number concentrations in spring and fall are lower (200 $cm^{-3}$) than in summer (430 $cm^{-3}$) and also in winter even though the total particle number concentration is lower in winter than in

the transition seasons (see SI Sec 3). The low $CCN_{0.2}$ number concentrations in spring and fall coincide with smaller particle sizes. In spring and autumn, new particle formation (NPF) events contribute substantially to the particle number concentration (Dal Maso et al., 2005). Those newly formed particles stay smaller than during summer because there are less VOC oxidation products available that would condense on the particles. However, these particles still have a rather high organic mass fraction, which makes them less hygroscopic. Thus, the $CCN_{0.2}$ and particle number concentrations are smaller

in spring and autumn compared to the summer (Paramonov et al., 2013; Petäjä et al., 2005). Note that while we refer to CCN at a supersaturation at 0.2 %, small particles could contribute to the CCN number concentration at higher supersaturations in which case the lower concentrations in spring and fall might not be as apparent. During summer, particles are larger on average with a $D_g$ of 70 nm, but have a similar hygroscopicity ($D_{crit}$ around 110 nm) to the spring and fall particles ($D_{crit}$ around 100 nm) owing to the larger fraction of organic aerosol components (compare Fig. 7). Nevertheless, more $CCN_{0.2}$ can be observed due to an increase in the overall particle number concentration likely owing to high pressure periods in which air masses from the south arrive carrying aged anthropogenic and biogenic particles. In the rain forest (ATT), concentrations are low during the rainy season ($< 500$ cm$^{-3}$) early in the year when natural aerosol sources dominate (China et al., 2016; Pöhlker et al., 2012; Wang et al., 2016) and higher during the dry season ($> 500$ cm$^{-3}$) as a result of in-basin transport of emissions from deforestation fires (Pöhlker et al., 2016). In the rainy season, the biogenic (natural) particles are also smaller ($D_g$ of 90 nm versus 130 nm in the dry season) and seem to be more hygroscopic with a $D_{crit}$ of about 100 nm. Seoul (SEO) is subject to monsoon influence in summer (June through September). However, in the urban environment the impact of the rainy season is not clearly visible, neither in the $CCN_{0.2}$ number concentration nor in the average particle size. This is likely due to the continuous emission of particles from urban sources. The variations of $D_{crit}$, $< 100$ nm in winter and $> 100$ nm in summer, seem to suggest that aerosol particles are more hygroscopic in winter than in summer, potentially due to changes in emission sources. At BRW, the influence of Arctic Haze (Barrie, 1986) is evident from the roughly a factor 5 higher $CCN_{0.2}$ number concentrations in late winter and spring with concentrations around 100 cm$^{-3}$. Also at FIK, the seasonal cycle is characterized by pollution events occurring in summer ($CCN_{0.2} > 500$ cm$^{-3}$) which are associated with long-range transport of biomass burning aerosol containing larger size particles and absence of precipitation (Bougiatioti et al., 2016). The coastal sites at the Atlantic (MHD) and Pacific (NOT) show relatively large variability in all measured parameters without exhibiting a distinct seasonal cycle. At MHD particles tend to be smaller in summer ($D_g$ around 70 nm). In summer, sea salt contributes less to the MHD aerosol particle population, which results in a smaller $D_g$. More sea spray in winter, because of higher wind speeds and wave breaking, explains the smaller $D_{crit}$ (70 nm versus 80 nm in summer) in that season (Yoon et al., 2007). At NOT, $CCN_{0.2}$ number concentrations seem to be lower in winter ($< 1000$ cm$^{-3}$) compared to other seasons ($> 1000$ cm$^{-3}$). This might be related to convection, cloud and precipitation formation, and hence wet particle removal, induced by the interplay of the cold winter monsoon and the warm currents at the ocean surface. The rural and continental background stations in Europe all show relatively flat seasonal cycles.

While the seasonal cycles inform how aerosol particle properties change over longer timescales, i.e., months, auto-correlation of the hourly $CCN_{0.2}$ number concentration time series can provide insights into the variability over shorter (synoptic) timescales, i.e., days. Figure 5 shows the persistence of $CCN_{0.2}$ number concentrations for winter (DJF) and summer (JJA). The persistence is essentially a metric for how long the pattern of CCN number concentrations "remains similar" (see Sec 2.3.1). A pattern can refer to a This does not exclude periodic variations on shorter time scales, such as diurnal cycle or simply an unvaried number concentrationcycles, than the observed persistence as long as the amplitude of

the periodic variations and the averages over the cycles remain similar. At MEL, CES and SMR, for example, the winter persistence is larger than five days, which is most likely related to the relatively stable weather patterns in winter when atmospheric blocking situations occur, which are anti-cyclonic, quasi-stationary high-pressure systems persisting for several days up to weeks that disturb the otherwise predominant westerly flow (Sillmann and Croci-Maspoli, 2009). Conversely, in

summer, persistence is only two days for MEL and CES reflecting likely a combination of the much more variable weather conditions and genuine changes in aerosol particle characteristics due to short- and medium-range transport, as well as intermittent new particle formation events (Manninen et al., 2010). Also, the amplitude of the boundary layer height between night and day is quite large introducing differences in particle concentrations due to dilution effects. At the mountain stations, the persistence is longer in summer. It is driven by the regularity of the boundary layer injections and the resulting

high particle number concentrations (Herrmann et al., 2015). It has to be noted that, in this case, the high persistence is an indication of a regular pattern rather than a constant $CCN_{0.2}$ number concentration. In the rain forest, the rainy season is characterized by a longer persistence (7.5 days) than the dry season (2 days) potentially owing to the regular rain events, i.e., similar to the boundary layer injections at the mountain stations. FIK shows higher persistence during summer (5 days) than winter (< 3 days), while the opposite is the case for all other coastal stations. At FIK weather patterns are stable in summer

and air masses originate from the N-NE sector for more than 80 % of the time (Kouvarakis et al., 2000). For VAV the longer persistence in summer (4.5 versus 2 days) as represented in this data set might reflect a peculiarity of the particular observation period. Generally, similar to SMR, CES and MEL, more stable conditions in winter are expected. The long persistence in winter at BRW (5.7 days) reflects the stable Arctic atmosphere which gets perturbed during spring and summer, when the Arctic Haze conditions fade. Note, since there was not enough data coverage for BRW in the summer

months, springtime (M,A) is shown. Persistence is low in SEO (1.2 days) and there is virtually no difference between seasons, likely due to the station's vicinity to emission sources that drive variability rather than synoptic patterns.

### 3.2 Activation

To compare the activation behavior of particles at all sites, we calculated the activation ratio (*AR*) for each measured SS based on the particle number size distribution > 20 nm. Further, to explore how the *AR* changes with SS, we form the ratio of *AR* at each SS (*AR_x*) to *AR* at SS = 0.5 % (*AR_{0.5}*). If CCN number concentrations were not measured at SS = 0.5 % the value was linearly interpolated. Results are shown in Fig. 6. Panel (a) shows all non-coastal sites, and panel (b) the coastal sites. The dashed black line represents a logarithmic fit through all curves following Eq. 6:

$$\frac{AR_x}{AR_{0.5}} = A * \ln(SS) + b \tag{6}$$

with $A = 0.46 \pm 0.02$ and $b = 1.31 \pm 0.02$. A steep slope means that the aerosol particle population activation is sensitive to small changes in the SS, while a flat slope indicates that a further increase in SS would not have a large impact on the *AR*.

The curves in panel (a) suggest that particles at all non-coastal sites, except for the rain forest location, have comparable activation properties with changing SS. This reflects the results shown in Fig. 3. These sites have similar ranges for the critical and geometric mean diameters. When fitting the average of the non-coastal curves, $A$ would be $0.54 \pm 0.01$ and $b = 1.41 \pm 0.01$. Particles observed in the rain forest follow the general non-coastal curve up to SS = 0.5 %. Thereafter, the curve flattens, meaning that the aerosol particle population is rather insensitive to higher SS and that most particles activate at SS $\leq$ 0.5 %. The frequency distribution of $D_g$ at ATTO (Fig. 3b) suggests that most particles are larger than 100 nm which will already activate at supersaturations lower than SS = 0.5 %. Regarding the lower activation ratio at higher SS, Pöhlker et al. (2016) link it to the influence of nearby biomass burning emissions and hence smaller less hygroscopic particles. Also, previous studies (e.g., Gunthe et al., 2009) confirmed this finding by showing that particles with an electrical mobility diameter < 90 nm are less hygroscopic than larger particles, owing to the difference in composition. The mass fraction of inorganic constituents is higher in larger particles.

The curves for the coastal sites exhibit more spread at both low and high SS (compare also Fig. 3). In the Arctic (BRW), for example, the curve suggests that most particles activate already at SS $\leq$ 0.3 %, which is in line with the measured large particles sizes and the observation that Arctic background aerosol particles are generally highly hygroscopic (Lathem et al., 2013). A similar observation is true for the Mediterranean environment. The observed activation behavior at MHD follows the average from all curves (dashed line) while particles at NOT are still sensitive to higher SS, similar to the "land-based" observations. This is most likely due to the influence from long-range transported anthropogenic air pollution at the site.

Overall it seems that at the coastal sites, the mixing between anthropogenic and natural (marine) sources leads to a complex behaviour of particle activation. Conversely, at continental sites the natural (biogenic) sources lead to size-distributions and hygroscopic characteristics that are comparable to the anthropogenic ones. For instance, NPF events supply ultrafine particles in place of combustion particles. As a consequence, very different places like JFJ, SMR, CES, MEL and SEO show similar geometric mean diameters and hence similar particle activation curves. For further details regarding the seasonal cycles of $AR$ we refer the reader to SI Sec 3.

## 3.3 Aerosol chemical composition and the composition-derived hygroscopicity parameter kappa

At seven stations, the aerosol particle chemical composition was measured by means of different types of aerosol mass spectrometers (see Table 1 for details). Figure 7 shows the seasonal cycle of inorganic and organic median mass concentrations on the left, and the evolution of $\kappa$ on the right throughout the year as median value and interquartile range. At most stations, nitrate plays a minor role except for the rural background stations CES and MEL, where it especially contributes during the colder months with up to 40 %. These two stations are closest to the central European high-$NO_x$ region (Beirle et al., 2004). The mass fraction of organics is mostly below 50 % at the two sites, and the hygroscopicity of the particles appears to be driven by the inorganic components, predominantly by ammonium nitrate: The larger the fractional contribution of nitrate ($f_{NO3}$), the higher $\kappa$ becomes: at CES $\kappa \geq 0.83\ x\ f_{NO3} + 0.11$ and at MEL $\kappa \geq 0.82\ x\ f_{NO3} +$

*0.12* (not shown). Note, that especially for the European sites it might be possible that a considerable fraction of nitrate is present in form of organic nitrate (Kiendler-Scharr et al., 2016), which is likely to influence the hygroscopicity. Similarly, particulate sulfate can be present as organosulfate (Vogel et al., 2016) in which case particle hygroscopicity would be overestimated. At all other stations, organics can play a more important role in terms of mass contribution (up to 80 % at

SMR, ATTO and JFJ, and up to 40 % at MHD and FIK) and determination of the $\kappa$ value. In the boreal forest, organics constitute the largest mass fraction throughout the year and especially during summer. In this season, the boreal forest is actively growing and producing more VOCs, whose oxidation products either condense on pre-existing particles or contribute to NPF events. Organic matter can dominate the particle composition, especially in the absence of long-range transport of other chemical constituents. In the rain forest (ATT), organic matter also dominates, contributing some 60 –

70 % to PM$_1$ throughout the year. Therefore, some of the observed hygroscopicity changes can be associated with differences in organic aerosol composition (i.e., its oxidation state), rather than differences in inorganic/organic fractions. At the high alpine site (JFJ) the influence of organic matter (up to 70 % mass contribution) becomes most important in summer because of boundary layer air mass uplift, and again the impact on the calculated $\kappa$ is evident. At the coastal sites in the Mediterranean (FIK) and Atlantic (MHD), the non-refractory submicron aerosol particle mass is driven by inorganic

components, predominantly sulfate (mass contribution of up to 50 %). However, increased organic particle mass is observed during the biomass burning season at FIK with 40 % mass contribution (Bougiatioti et al., 2016), when $\kappa$ reaches a minimum, and in springtime at MHD (also 40 %), as has been observed previously (Ovadnevaite et al., 2014). At MHD, $\kappa$ is generally > 0.5 owing to the influence of sea salt, but at the same time is also very variable (0.45 to 0.92 in the monthly median) owing to the mixed influences of marine organic aerosol and anthropogenic air pollution.

Figure 8 provides further indication on how the CCN number concentration is related to the aerosol particle mass and chemical composition. Binned averages and standard deviations of inorganic (ammonium, nitrate, sulfate, chloride, and sea salt) and organic particle mass are shown against CCN$_{0.2}$ number concentrations. Bins represent 50 particles cm$^{-3}$. The solid lines are the linear fits through inorganic and organic mass concentration data with all parameters indicated in the table. Generally, the correlation between particle mass and CCN$_{0.2}$ number concentration is high and similar for organic and

inorganic components (R > 0.81 for all cases except for inorganics at SMR where R = 0.66). The similarity might be an indication for internally mixed particles or the co-existence of different particles types at the observatories. At CES, the CCN number concentration is more strongly influenced by the inorganic aerosol particle mass, as can be concluded from the higher correlation coefficient compared to the one of CCN$_{0.2}$ number concentration and organic particle mass (R = 0.93 versus 0.86). At FIK, the correlation coefficient with inorganics is only slightly higher (0.97 versus 0.94), while at MEL,

MHD, and JFJ they are roughly equal. This relates to the average over the whole year, while seasonally there can be significant differences, as Fig. 7 shows. In the forest environments, correlations of CCN number concentrations with organic particle mass are higher than for inorganic particle mass (0.94 versus 0.89 at ATTO, and 0.97 versus 0.66 at SMR). From this perspective, it is clear that knowing the share of organic particle mass is important for understanding the activation behavior of the specific particle population at each site.

A negative relationship of the composition-derived $\kappa_{IA+OA-BC}$ value and the ratio of organic to inorganic particle mass can be observed as shown in Fig. 9. Generally, the curve follows a two component system that can be described by Eq. 2 with $i$ standing for the inorganic and organic aerosol components. The figure indicates how well $\kappa$ can be described when knowing the organic to inorganic aerosol ratio. The spread in $\kappa$ values between locations, especially at lower ratios, is due to the heterogeneity in the composition of the inorganic particle components. For example, at CES and MEL ammonium nitrate constitutes a large fraction of the inorganic aerosol mass, while at ATT and SMR particulate sulfate such as salt or acid dominates. However, the vertical distance in the lines for ATT and SMR shows that it makes a significant difference whether sulfate is present as sulfuric acid ($\kappa = 0.73$) or as ammoniumsulfate ($\kappa = 0.6$). For SMR, similar observations have been made investigating the relationship of the organics-to-sulfate ratio to the *GFs* for certain particle sizes (Hong et al., 2014). For higher ratios, $\kappa$ values from all stations converge when assuming one single hygroscopicity for OA, i.e., $\kappa_{org} = 0.1$, because $\kappa_{org}$ starts to dominate the result. Note that the asymptotic-like approach of the curves towards a certain $\kappa$ value cannot be interpreted as $\kappa_{org} > 0.1$ for that reason.

## 3.4 Closure study

Achieving closure between measured and predicted CCN number concentrations has been tried in a large number of studies reflecting conditions in a variety of environments such as cities, high alpine stations, and boreal, tropical, and mid-latitude forests etc. (e.g., Almeida et al., 2014; Asmi et al., 2012; Hong et al., 2014; Jurányi et al., 2010; Kammermann et al., 2010b; Pöhlker et al., 2016; Wu et al., 2013). Most of these studies, however, rely on relatively short data sets from days to several weeks at most. Ervens et al. (2010) present an overview of closure studies from six different sites and an extensive comparison with other studies discussing the influence of the particles' mixing state and the hygroscopicity of the organic fraction, as well as the distance from emission sources. Generally, they find that ratios of predicted over measured CCN number concentrations can range from 0.2 to 7.9, with results further away from emission sources becoming more reliable. This observation has been confirmed, for example, by closure studies at the high alpine sites, which are relatively far away from emission sources (Asmi et al., 2012; Jurányi et al., 2010). However, other studies suggest that poor performance of closure studies near sources can likely be attributed to difficulties in measuring the relevant aerosol properties with sufficient resolution in time and at relevant particle sizes, rather than to intrinsic limitations of the applied $\kappa$-Köhler theory (Jurányi et al., 2013). Ervens et al. (2010) suggest that organic particle matter can be treated as hygroscopic (they use $\kappa_{org} = 0.12$) a few tens of kilometers downwind from emission sources. With this $\kappa_{org}$ value and varied assumptions about aerosol particle hygroscopicity and state of mixing - that can lead to similar results due to compensating effects - reasonable closure within a factor of two can be achieved, even though the true nature of the aerosol particle population is not known. Jurányi et al. (2010) also show that uncertainties in the bulk $\kappa$ value can lead to only a factor of two difference between measurement and prediction at low SS and even less at high SS. Larger discrepancies hence suggest that either the classical $\kappa$-Köhler theory

does not hold, e.g., because of the particles' surface tension (e.g., Ovadnevaite et al., 2017), kinetic limitations, or other reasons, or, which is mostly the case, that there are issues with the measured data of particle number concentration, size distribution, and CCN number concentrations (see SI Sec 2).

Based on these previous results and the fact that all stations with available chemical composition data are at least 20 km away from large emission sources, we performed simple closure studies assuming internal mixtures and a $\kappa_{org}$ value of 0.1. We focus on the long-term performance of the instruments that were run in monitoring mode, implying less attendance than during intensive field campaigns, and the sensitivity of the results to changes in the assumptions:

1. varying the approach to translate composition measurements to $\kappa$ values as given in Table 2,
2. applying a fixed shape of the particle number size distribution (the average of the entire data set) while keeping the total number concentration of particles temporally variable as measured and applying $\kappa_{IA+OA\text{-}BC}$, and
3. applying the temporally variable particle number size distribution, but scaled to the median particle number concentration as measured at each station with $\kappa_{IA+OA\text{-}BC}$.

This approach is similar to the one shown by Jurányi et al. (2010) in their Fig. 6, focusing on a one month data set at JFJ. Within this study, however, closure performance of seven stations over at least one year can be compared.

The results are shown in Fig. 10a for SS = 0.5 % with the correlation coefficient of predicted over measured particle number concentrations on the vertical axis and the geometric mean of the particle number concentration ratio on the horizontal axis. We use the geometric instead of the arithmetic mean, because particle and CCN number concentrations are log-normally distributed. This can result in slightly different mean values compared to the arithmetic mean, which has been used more frequently in previous studies (e.g., Ervens et al., 2010). Table 3 provides a comparison of both means. The correlation coefficient is a measure of the agreement between instruments over time, i.e., the stability of instrumental performance. The ratio of the predicted and measured $CCN_{0.5}$ number concentrations indicates the quality of the average prediction with 1 being a perfect prediction and numbers < 1 (> 1) being an under (over) prediction. Looking only at closure results with $\kappa_{IA+OA\text{-}BC}$ and $\kappa_{IB+OA\text{-}BC}$ predictions fall within a range of ratios between 0.87 and 1.37, which qualifies as rather good agreement compared to the findings in the overview by Ervens et al. (2010) but reflect a similar range of results as described by Kammermann et al. (2010b) based on hygroscopicity tandem DMA studies. Values for $R$ fall between 0.87 and 0.98, i.e., the accuracy of predicting temporal variability is high. This means for this particular selection of stations that only the average bulk hygroscopicity of the particles needs to be known to obtain a realistic estimate of the CCN number concentration. Data for the CES observatory are located in the area of over-prediction between a factor of 2.5 and 3.1 due to losses of small particles in the aerosol sampled by the CCNC (see Sec. 2.2 and SI Sec. 1 for more details). Results are shown nevertheless for completeness. Including BC concentrations in the calculation of $\kappa$ has limited influence on the overall closure performance, not enlarging the range of predicted vs measured data. This means that for long-term observations neglecting the BC mass concentrations has only a limited effect at such types of sites. Slight variations in the chemical composition and with that in the aerosol particle hygroscopicity only play a minor role for the accurate prediction of $CCN_{0.5}$ number concentrations that fall within a factor of two for this data set. This has been expressed in a number of previous

studies (e.g., Dusek et al., 2006; Juranyi et al., 2011; Juranyi et al., 2010; Pohlker et al., 2016). Even a fixed $\kappa$ of 0.3 can represent the aerosol particle hygroscopicity sufficiently well for CCN predictions, with a range of 0.82 to 1.38 for the ratio of predicted over measured $CCN_{0.5}$ number concentrations. A $\kappa$ of 0.3 has been suggested earlier to be generally representative of polluted continental environments (Andreae and Rosenfeld, 2008). This also seems to hold for other environments that partly represent free tropospheric conditions (JFJ) and the Amazon rain forest conditions in the dry and rainy season including natural forest emissions and long-range transport of Amazonian and African biomass burning aerosol pollution, as well as Saharan dust (ATT). Coastal sites (MHD, FIK) can also be represented by the same $\kappa$ value. However, this value is too high for the city in East Asia (SEO).

An influence on the closure results is also observed, when the shape of the particle number size distribution is fixed, but scaled to the measured particle number concentration at each site (dark blue symbols in Fig. 10a). The predictability of averaged $CCN_{0.5}$ number concentrations decreases moderately for all stations (except CES), and is within the boundaries of the ratio of 0.80 and 1.96. However, the correlation between the predicted and measured CCN number concentration naturally decreases as the fixed shape of the particle number size distribution does not represent the changing number fraction of particles with diameters larger than $D_{crit}$ over time. The correlation coefficient drops more strongly for the MEL and SMR, which is due to the regular presence of a large numbers of small particles at these observatories due to new particle formation events (Birmili and Wiedensohler, 2000; Dal Maso et al., 2005; Manninen et al., 2010). The relatively large fraction of small particles can be seen in Fig. 3b expressed as the $D_g$ frequency. The fixed shape of the particle number size distribution represents these two stations least accurately.

Keeping the number concentration of particles fixed at each station's median and scaling the temporally variable particle number size distribution to it, generally results in the poorest predictability (ratios between 0.65 and 2.28). The temporal prediction skills drop to correlation coefficients < 0.7 for all stations as the temporal variability in the data set is mostly driven by changes in particle number concentrations. This is especially true for MHD, where the correlation coefficient is as low as 0.2, because the particle concentrations are more variable at this location than at any other one (see Fig. 4 in Schmale et al. (2017)).

Applying these observations to the stations without aerosol chemical particle composition measurements, we performed closure studies at SS = 0.5 % based on a calculated average $\kappa$ value per site category: rural background, $\kappa = 0.48$ from MEL and CES; PUY: alpine, $\kappa = 0.41$ from JFJ; BRW and NOT: coastal background, $\kappa = 0.55$ from MHD and FIK. For the urban station, SEO, we use $\kappa = 0.1$ (Schmale et al., 2017). In addition, $\kappa = 0.3$ is applied to all stations. Results are shown in Fig. 10b. CCN number concentrations can be reproduced within 1.02 and 1.99 for the category-averaged $\kappa$ values and within 1.03 and 1.75 for $\kappa = 0.3$. For NOT the averaged $\kappa$ value is representative, likely because of the mixture of the highly hygroscopic sea salt and sulfur-rich marine accumulation mode particles with the local aerosol populations. At BRW, the Arctic coastal environment, particles seem slightly less hygroscopic, leading to better results with $\kappa = 0.3$ rather than 0.55. For SEO, the urban $\kappa$ value is also better suited than the suggested global average of 0.3, while for PUY there is only a small difference between the alpine and global average $\kappa$ values. At VAV, the rural background $\kappa$ value is too high, leading to a significant

over-prediction by a factor of two. In the previous estimate at the rural continental site VAV by Paramonov et al. (2015) $\kappa$ values are around or below 0.3 depending on dry particle diameter, which are closer to the $\kappa$ values presented in Table 2 at the forest station SMR. This is not surprising since the size distributions at VAV and SMR are similar (Fig. 3) and VAV is also a northern station, and is surrounded by forest regions similar to SMR. Furthermore, it is possible that particulate nitrate and sulfate at CES and MEL were associated with organic matter in which case the hygroscopicity of the particles would be overestimated even though results in Fig. 10a do not suggest so. Hence, care must be taken when choosing representative $\kappa$ values. Two stations in the same site category could have $\kappa$ values that are actually significantly different (compare the forest stations in Fig. 9), and two stations in two different site categories could have similar $\kappa$ values.

In general, the correlation coefficients range between 0.70 and 0.93 for site category specific $\kappa$ values and for an invariant $\kappa$ value of 0.3. Given that these $\kappa$ values do not reflect the temporal variability of the chemical composition at the stations, the prediction accuracy is reasonably high.

Other than the varied parameters shown in Fig. 10a, the value of the surface tension of the solution in the droplet might play a role. Based on JFJ data, using the closure calculations with $\kappa_{IA+OA-BC}$, a 30 % decrease (increase) in $\sigma_{sol}$ would result in a 17 % under-prediction (over-prediction of 25 %, see SI Sec 2) of $CCN_{0.5}$. This is within the range of change introduced by fixing the particle number concentration or size distribution. However, such a large change in $\sigma_{sol}$ is not likely as a 30 % decrease can happen if very strong surfactants are present (Petters and Kreidenweis, 2013). Furthermore, small errors in determining the measured instrument supersaturation will have very little influence on the ratio of predicted versus measured CCN number concentrations, i.e., roughly 5 % when misrepresenting SS by an assumed 10 % (see SI Sec 2). Based on this, determining the particle number concentration and size distribution as precisely as possible is most important for the successful prediction of CCN number concentrations at regionally representative observatories in all regions studied here. For model simulations, this means that it should be sufficient to represent the particle number concentration and size distribution correctly and roughly the chemical composition. However, it remains to be shown whether this is true for other stations not studied here.

## 4 Summary and Conclusions

We have analyzed long-term data from collocated measurements of CCN number concentrations, particle number size distributions, and in some cases submicron aerosol chemical composition from different regions.

### Regional variability

It is evident that CCN number concentrations vary considerably with region. However, there are only a few long-term studies that have compared number concentrations from the same station category across different regions. Previous model studies (Pringle et al., 2009) have investigated the effect of applying particle number size distribution data representative of

one region to another when attempting to predict the number of cloud droplets, and found that errors can be as large as 75 % in the high latitudes and in regions with persistent stratocumuli. Even though the number of stations is limited to twelve, this study comprises sites from Europe, the Americas, and Asia with four stations representing coastal background, three stations rural background, two alpine sites, two forest sites, and one urban location. Our results (Figs. 1b, 3 and 4) show that $CCN_{0.2}$

number concentrations do not only vary considerably by region but also within one station category, e.g., by up to a factor of 30 in spring among the coastal stations between the Arctic and Asian Pacific, or by up to a factor of four in spring among the rural background stations. The alpine stations exhibit differences around a factor of two, while the two particular forest environments are relatively similar despite representing high and tropical latitudes. In terms of particle activation behavior, Fig. 6 shows that, while most non-coastal stations exhibit similar characteristics, the Amazon rain forest is different, and

there is a relatively large spread among the coastal stations. This demonstrates that a broad regional data coverage is necessary to understand the actual variability of $CCN_{0.2}$ number concentrations especially for coastal sites.

**Seasonal variability**

$CCN_{0.2}$ number concentrations follow a seasonal cycle at most stations (Figs. 4 and 5). This means that short-term

measurements can only be representative of the season in which they were performed. A comparison with data from the short-term EUCAARI dataset relying on comparable measurement protocols (Paramonov et al., 2015), covering three of the stations discussed here for a short duration, shows significant differences in the $CCN_{0.2}$ number concentrations. At CES, this study's average concentration is four times higher than the EUCAARI summer 2008 data. In the Amazon, the winter 2008 average represents only 10 % of the annual average covered here; and at FIK, the summer through fall observations in 2007

covering the biomass burning season result in an average concentration that is twice as high as the full year 2015 measurements. Comparing our data with EUCAARI data covering one or more years and not overlapping with our observation period at JFJ, SMR and VAV, results in discrepancies no larger than a factor of 1.3, and for MHD in a factor of 2. This means that the long-term observations covered in this study are largely representative for those sites, however, inter-annual variability can still lead to differences in concentrations. Looking at $CCN_{0.2}$ number concentration persistence, all

stations, except the urban environment, show marked differences between winter and summer. This indicates as well that short-term observations cannot be extrapolated over seasons, an important aspect to keep in mind when comparing model results with observations.

**Prediction of $CCN_{0.5}$ number concentrations**

From the closure studies, we learn that when applying a simple κ-Köhler formulation assuming internal mixture and size-independent particle hygroscopicity, the geometric mean ratio between predicted and measured $CCN_{0.5}$ number concentrations end up in the range between 0.87 and 1.37. The ratio exhibits a high reproducibility of temporal variability reflected by statistically significant correlation coefficients between 0.87 and 0.98. This prediction accuracy is rather high compared to previous synthesis studies that found a range between 0.2 and 7.9 (Ervens et al., 2010), potentially owing to the

relatively remote location of the observatories discussed here and the apparently high data quality. These results were obtained by using the ion composition to derive $\kappa$ for inorganic aerosol constituents, while $\kappa_{org}$ was assumed to be 0.1 and no information on BC mass concentrations was used. Assuming $\kappa_{org} = 0.1$ worked sufficiently well in the present study, as the OA contribution to the submicron aerosol mass is mostly below 50 %, except at the forest sites, where it is higher. In the latter case, however, $\kappa_{org} = 0.1$ still seems to be a reasonable approximation. Pöhlker et al. (2016) determined an effective $\kappa_{org}$ of 0.12 for the Amazon rainforest. When assuming an overall $\kappa = 0.3$, similarly good agreement between measured and predicted $CCN_{0.5}$ number concentrations is obtained.

Sensitivity studies show that the temporal variability of $CCN_{0.5}$ number concentrations would be poorly represented with an unknown actual particle number concentration, i.e., the correlation coefficient drops below 0.7 for all stations and as low as 0.2 for MHD. Also an invariant particle number size distribution can lead to very low correlation coefficients of $< 0.35$ for some stations. This means that temporally resolved data of particle number concentration and their size distribution are essential to predict $CCN_{0.5}$ number concentrations. Conversely, a fixed $\kappa$ value does not significantly reduce the correlation coefficients but influences the $CCN_{0.5}$ number concentration predicted on average (Fig. 10). Care must be taken when applying station type averaged $\kappa$ values to stations of the same category without chemical observations. While on average the prediction accuracy lies within a factor of 1.36, for individual stations the overestimation can be as large as 200 %, in this case of VAV. VAV belongs to the rural background site category, which apparently is not suitable for VAV in terms of predicted $CCN_{0.5}$ number concentrations from site category specific $\kappa$ values. Namely, VAV $\kappa$ values are more similar to the values at the forest station category.

**General implications**

The potential CCN number concentration alone cannot determine the actual cloud droplet number concentration (CDNC), the variable that is important to describe cloud radiative properties. Other factors such as the updraft velocity and the resulting water vapor supersaturation, at which particles are activated play an important role (Reutter et al., 2009). The CCN-limited regime applies to lower CCN number concentrations of, e.g., less than 9000 cm$^{-3}$ for SS = 0.2 % and a $\kappa$ value around 0.4 which is roughly representative of this data set. This means that all stations considered here would fall into the CCN-limited regime, except for SEO occasionally. Against this background and given the results of the closure studies performed here with $\kappa_{IA+OA-BC}$, CCN number concentration predictions are within the range of roughly $\pm$ 30 % for stations with aerosol particle chemical composition information. Based on Sotiropoulou et al. (2006), who found that errors in CCN prediction result at most in half the error for CDNC, we find that CDNC can be predicted within $\pm$ 15 % from data collected at regionally representative observatories. Similarly, Moore et al. (2013) found a CDNC sensitivity of 10 – 30 % to CCN abundance over the continents, which would further reduce the uncertainties of CDNC predictions based on this data set. Considering our results for stations without particle chemical observations, CCN number concentrations are overestimated on average by 36 %, leading to CDNC overestimation of $\leq$ 18 %. However, at individual stations like VAV, the CCN number concentration is overestimated by a factor of two in our closure experiments which would result in an overestimation

of $\leq 50$ % of the CDNC. Such a misrepresentation would result in precipitation underestimation for locations with shallow cloud formation, as precipitation efficiency in shallow convection is reduced with increasing CDNC (Andreae and Rosenfeld, 2008; Rosenfeld, 2000).

**Recommendations**

Given that operating extensive equipment for aerosol particle characterization is expensive and labor intensive, it will not be possible to undertake the same observational efforts as discussed here at many stations across the globe. However, information of the CCN number concentration in many locations is important for modeling aerosol cloud interactions more 10 accurately and to constrain their radiative forcing better. Based on this study, we can recommend that observations of particle number size distributions at regionally representative sites would be sufficient when CCN number concentration measurements are run in parallel for the duration of at least one year. From the collocated observations, a temporally-resolved $\kappa$ value based on the simple formulation of the $\kappa$–Köhler theory can be derived and applied to the particle number size distribution to derive the CCN number concentration once the direct measurements have been concluded. This avoids 15 operational expenses from sustained operation of a CCNC as well as from instruments capable of producing highly time-resolved aerosol chemical composition data. This statement is, however, only applicable to the context of investigating aerosol-cloud interactions as discussed here. Chemical composition measurements are indispensable in other contexts, e.g., when studying air quality. Furthermore, suggesting to find an alternative to measuring highly time-resolved particle chemical composition is not to say that such data are not desirable, especially because they allow for source apportionment studies that 20 can provide results highly valuable to interpret CCN number concentrations (e.g., Bougiatioti et al., 2016). In the ACI context, not using composition-derived $\kappa$ values also circumvents added uncertainty from the measured aerosol chemical component concentrations and the bias towards the mass size distribution maximum. With respect to monitoring only particle number size distributions and applying a critical diameter to derive CCN number concentrations, a study for JFJ confirms that such an approach is reasonable; Hoyle et al. (2016) showed that 79 % of the variance in cloud droplet number 25 concentration can be explained by the CCN number concentration based on a $D_{crit}$ of 80 nm. Based on the suggested simplified measurement strategy together with our observation of high CCN number concentration variability within site categories, it is conceivable to operate several "migrating CCNCs" around the world where long-term particle number size distribution data are already available. These CCNCs would have to be calibrated regularly at the World Calibration Center for Aerosol Physics in Leipzig, Germany, to assure data quality (http://actris-ecac.eu/reports.html). 30 Last but not least, we encourage the modeling community to make use of this data set to evaluate CCN results near the observatories and discuss the simulation skills of the models, and to provide recommendations for priority observation sites where our simplified measurement recommendation can be employed.

**5 Data availability**

All data are available from: http://actris.nilu.no/Content/products; data for the ATTO station have been submitted as supplementary material to Pöhlker et al. (2016).

## 7 Supplementary link

## 8 Author contribution

JS, MG, UB, and FS devised the study. JS analyzed the data and wrote the manuscript. SH, SD, BH, HK, KS, JO, JB, AB, AK, NK, IS, SC, AJ, MP, PS, YI, UP, LP, AM, SY, UB, MG contributed to the data interpretation. PA, MÄ, NB, ME, GF, RF, AF, EH, HH, RH, GK, MK, NM, AN, CD, TP, DP, CP, AP, ES, MA, PA, AW, JO provided detailed information on the data sets. All authors were involved in the data acquisition and commented on the manuscript.

## 9 Competing interests

The authors declare no competing interests.

## 10 Disclaimer

## 11 Acknowledgements

The authors acknowledge funding from the European FP7 project BACCHUS (grant agreement No. 49603445) and the Horizon 2020 research and innovation programme ACTRIS-2 Integrating Activities (grant agreement No. 654109). Long-term measurements at Jungfraujoch are supported by the International Foundation High Altitude Research Stations Jungfraujoch and Gornergrat and MeteoSwiss in the framework of the Global Atmosphere Watch (GAW) program. Measurements at Mace Head are supported by HEA-PRTLI4 Environment and Climate: Impact and Responses programme and EPA-Ireland. The research at Cabauw has received funding from the European Union Seventh Framework Programme (FP7, grant agreement no. 262254). We appreciate the support from KNMI in hosting the experiment at Cabauw and for the access to meteorological data from the tower. We also thank Philip Croteau (Aerodyne Research) for his support on the Cabauw ACSM measurements regarding the data acquisition and evaluation. The research at Noto was supported by JSPS Grant-in-Aid for Young Scientists (A, grant number JP26701001). The research in Seoul was supported by Grant KMIPA 2015–1030. For the operation of the ATTO site, we acknowledge the support by the German Federal Ministry of Education and Research (BMBF contract 01LB1001A) and the Brazilian Ministério da Ciência, Tecnologia e Inovação (MCTI/FINEP contract 01.11.01248.00) as well as the Amazon State University (UEA), FAPEAM, LBA/INPA and SDS/CEUC/RDS-Uatumã. This paper contains results of research conducted under the Technical/Scientific Cooperation Agreement between the National Institute for Amazonian Research, the State University of Amazonas, and the Max-Planck-Gesellschaft e.V.; the opinions expressed are the entire responsibility of the authors and not of the participating institutions. We highly acknowledge the support by the Instituto Nacional de Pesquisas da Amazônia (INPA). We thank the Max Planck Society (MPG) and the Max Planck Graduate Center with the Johannes Gutenberg University Mainz (MPGC). The research at

Vavihill has been supported by the Swedish research councils VR and FORMAS and the strategic research area MERGE. The research at Hyytiälä was supported by the Academy of Finland Center of Excellent Programme (grant no. 307331). We thank all observatories' operational teams for their continuous efforts. AN thanks the Georgia Power faculty Chair and Cullen-Peck faculty fellowship funds.

**Tables**

**Table 1: List of measurement sites participating in this synthesis study. Station names followed by an asterisk (*) are part of the ACTRIS network. Abbreviations correspond to those within the Global Atmosphere Watch network/programme.**

| | Station name | ATTO | Barrow | CESAR Tower | Finokalia | Jungfraujoch | Melpitz |
|---|---|---|---|---|---|---|---|
| station information | abbreviation | ATT | BRW | CES | FIK | JFJ | MEL |
| | country | Brazil | Alaska, USA | The Netherlands | Northern, Crete, Greece | Switzerland | Germany |
| | coordinates | 02°07'S, 58°60'W | 71°19'N, 156°37'W | 51°58' N, 04°56'E | 35°20'N, 25°40'E | 46°33'N, 07°59'E | 51°32'N, 12°56'E |
| | elevation | 130 | 11 | -1 | 250 | 3580 | 86 |
| | site category | rainforest | Arctic maritime, coastal | near coast, rural back-ground | coastal background, Mediterranean | high alpine, background | continental background |
| CCN measurements | instrument type | DMT CCN-100 | DMT CCN-100 | DMT CCN-100 | DMT CCN-100 | DMT CCN-100 | DMT CCN-100 |
| | time coverage | Mar 2014 - Feb 2015 | Jul 2007 - Jun 2008 | Oct 2012 - Apr 2014 | Nov 2014 - Sep 2015 | Jan 2012 - Dec 2014 | Aug 2012 - Nov 2014 |
| | operation mode | monodisperse | polydisperse | polydisperse | polydisperse | polydisperse | monodisperse |
| | supersaturations (%) | 0.11, 0.15, 0.20, 0.24, 0.29, 0.47, 0.61, 0.74, 0.90, 1.10 | 0.20, 0.30, 0.50, 0.60, 1.00, 1.20, 1.45 | 0.10, 0.20, 0.30, 0.50, 1.00 | 0.20, 0.40, 0.60, 0.80, 1.00 | 0.10, 0.15, 0.20, 0.25, 0.30, 0.35, 0.40, 0.50, 0.70, 1.00 | 0.10, 0.20, 0.30, 0.50, 0.70 |
| size distribution measurements | instrument type | SMPS, TSI 3080 | TROPOS-type custom-built SMPS | SMPS TSI 3034 | TROPOS-type custom-built SMPS | Custom-built SMPS | TROPOS-type Dual SMPS custom-built |
| | time coverage | Mar 2014 - Feb 2015 | Sep 2007 - Jun 2008 | Jan 2012 - Dec 2014 | Nov 2014 - Sep 2015 | Jan 2012 - Dec 2014 | Jan 2012 – Jun 2014 |

| | | | | | | | |
|---|---|---|---|---|---|---|---|
| | diameter range (nm) | > 9 - 445 | 10 – 810 | 10 – 516 | 9 – 849 | 20 – 600 | 5 – 800 |
| chemical composition measurements | instrument type | Q-ACSM | | Q-ACSM | Q-ACSM | ToF-ACSM | Q-ACSM |
| | time coverage | Mar 2014 - Feb 2015 | | Jul 2012 - May 2013 | Sep 2014 - Sep 2015 | Jul 2012 - Jul 2013 | Jun 2012 - Jun 2014 |
| | species | ammonium, chloride, nitrate, organics, sulfate | | ammonium, chloride, nitrate, organics, sulfate | ammonium, chloride, nitrate, organics, sulfate | ammonium, nitrate, organics, sulfate | ammonium, chloride, nitrate, organics, sulfate |
| | collection efficiency | 1.0 (Jan-Jul); 0.5 (Aug-Dec) | | based on Mensah et al., (2012) | 0.5 | 1 | based on Middlebrook et al. (2012) |

| | Station name | Mace Head | Noto Peninsula | Puy de Dôme | Seoul | Smear | Vavihill |
|---|---|---|---|---|---|---|---|
| station information | abbreviation | MHD | NOT | PUY | SEO | SMR | VAV |
| | country | Ireland | Japan | France | South Korea | Finland | Sweden |
| | coordinates | 53°20'N, 09°54'W | 37°27'N 137°22'E | 45°46'N, 02°57'E | 37°34'N 126°58'E | 61°51'N, 24°17'E | 56°01'N, 13°09'E |
| | elevation | 5 | 0 | 1465 | 38 | 181 | 172 |
| | site category | coastal background | coastal background | mountain, continental background | urban, monsoon influence | rural background, boreal forest | rural background |
| CCN measurements | instrument type | DMT CCN-100 | DMT CCN-100 | mini-CCNC | DMT CCN-100 | DMT CCN-100 | DMT CCN-100 |
| | time coverage | Jul 2011 - May 2012 | May 2014 - Feb 2015 | Nov 2014 - Sep 2015 | Oct 2006 - Dec 2010 | May 2012 - Dec 2014 | Dec 2012 - Nov 2014 |

| | | | | | | |
|---|---|---|---|---|---|---|
| | operation mode | polydisperse | monodisperse | monodisperse | polydisperse | polydisperse | polydisperse |
| | supersaturations (%) | 0.10, 0.25, 0.35,  0.50, 0.75, 1.00 | 0.10, 0.20, 0.50, 0.80 | 0.2 | 0.20, 0.40, 0.60, 0.80 | 0.10, 0.20, 0.30, 0.50, 1.00 | 0.10, 0.15, 0.20, 0.25, 0.30, 0.35, 0.40, 0.50, 0.70, 1.00, 1.40 |
| size distribution measurements | instrument type | custom-built SMPS | DMA: TSI Model 3081L, CPC: TSI Model 3776 | Custom-built DMPS | TSI SMPS 3936L10 | UHEL-type custom-built Dual DMPS | ULUND-type custom-built Dual-DMPS |
| | time coverage | Jan 2011 - Dec 2012 | May 2014 - Feb 2015 | Nov 2014 - Sep 2015 | Jan 2006 - Dec 2010 | Jan 2012 - Jun 2014 | Dec 2012 - Nov 2014 |
| | diameter range (nm) | 25 – 500 | 8 – 342 | 10 – 400 | > 10 – 478 | > 3 – 1000 | > 3 – 900 |
| chemical composition measurements | instrument type | HR-ToF-AMS | | | | Q-ACSM | |
| | time coverage | Jan 2011 - Dec 2012 | | | | Mar 2012 - Sep 2013 | |
| | species | ammonium, chloride, nitrate, organics, sulfate, sea salt | | | | ammonium, chloride, nitrate, organics, sulfate | |
| | collection efficiency | based on Middlebrook et al. (2012) | | | | 0.52 | |

**Table 2: Median values (based on all data) for the bulk particle composition-derived hygroscopicity parameter kappa ($\kappa$) at each station with particle chemical composition measurements. The subscripts to $\kappa$ mean: '*IA+OA-BC*' inorganic aerosol and organic aerosol mass but no black carbon were considered; '*IB+OA-BC*' ion balanced inorganic aerosol and organic aerosol mass but no black carbon; '*IA+OA+BC*' inorganic aerosol, organic aerosol mass and black carbon; '*IB+OA+BC*' ion balanced inorganic aerosol, organic aerosol mass and black carbon were considered.**

| station | $\kappa_{IA+OA-BC}$ | $\kappa_{IB+OA-BC}$ | $\kappa_{IA+OA+BC}$ | $\kappa_{IB+OA+BC}$ |
|---------|------|------|------|------|
| ATT | 0.26 | 0.21 | 0.25 | 0.20 |
| CES | 0.52 | 0.50 | 0.50 | 0.48 |
| FIK | 0.48 | 0.47 | 0.46 | 0.45 |
| JFJ | 0.41 | 0.31 | 0.39 | 0.29 |
| MEL | 0.43 | 0.42 | 0.42 | 0.42 |
| MHD | 0.63 | 0.63 | 0.61 | 0.61 |
| SMR | 0.30 | 0.29 | 0.27 | 0.25 |

**Table 3: Comparison of geometric to arithmetic mean values of the ratios of predicted and measured CCN$_{0.5}$ number concentrations based on calculations with the composition-derived $\kappa_{IA+OA-BC}$.**

| station | geometric mean | arithmetic mean |
|---------|------|------|
| ATT | 1.06 | 0.94 |
| CES | 3.10 | 2.31 |
| FIK | 0.87 | 0.84 |
| JFJ | 1.09 | 0.93 |
| MEL | 1.23 | 1.28 |
| MHD | 1.14 | 1.14 |
| SMR | 1.32 | 1.19 |

Figures

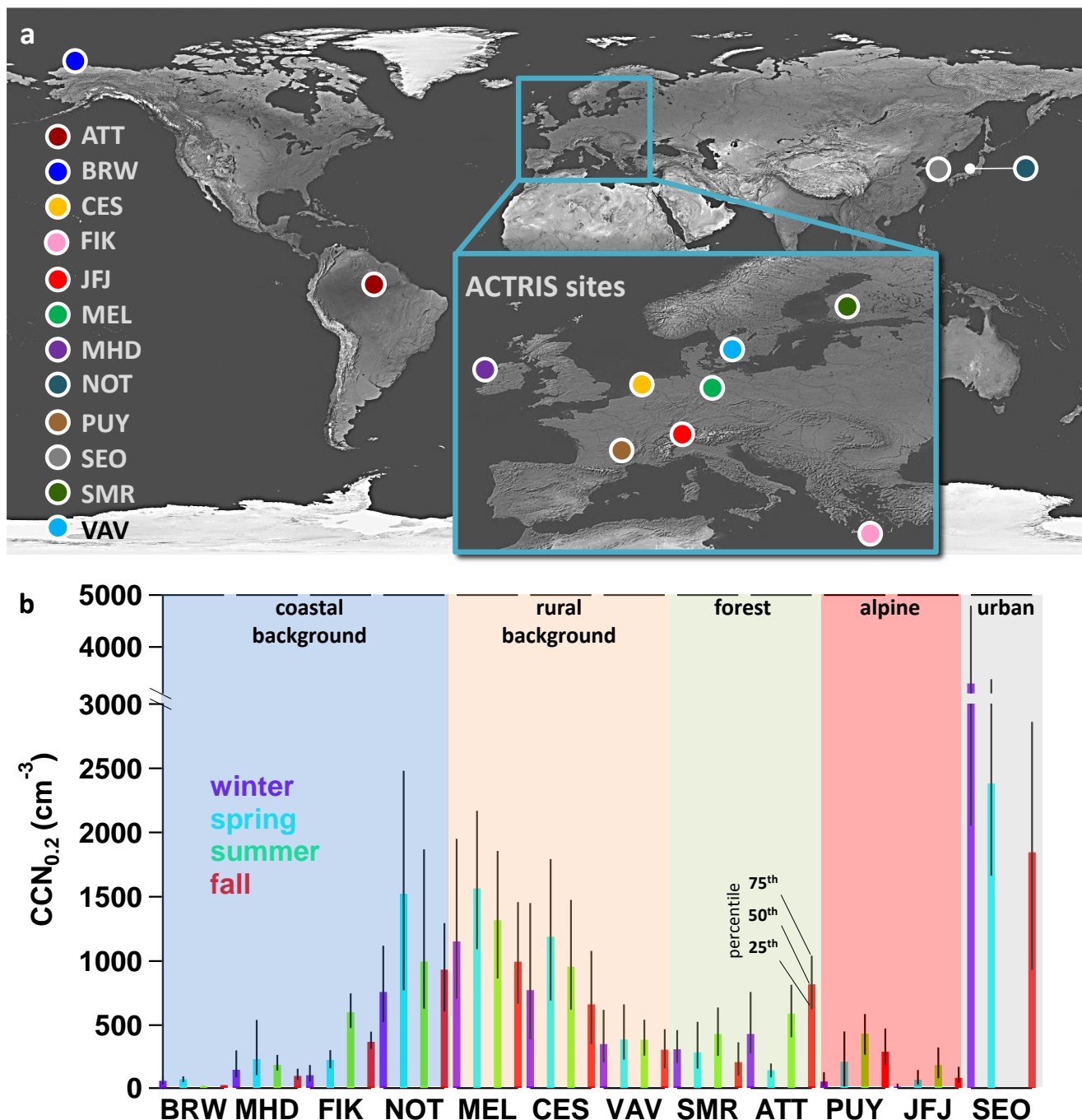

**Fig. 1**: (**a**) Map showing all measurement sites. Station abbreviations are given in Table 1. All stations in Europe are part of the ACTRIS
network. This map is adapted from Natural Earth III and Schmale et al. (2017). (**b**) Median and interquartile ranges of the seasonal CCN

number concentrations at a supersaturation of 0.2 % are displayed for each station. The shaded areas group the stations into the classifications indicated.

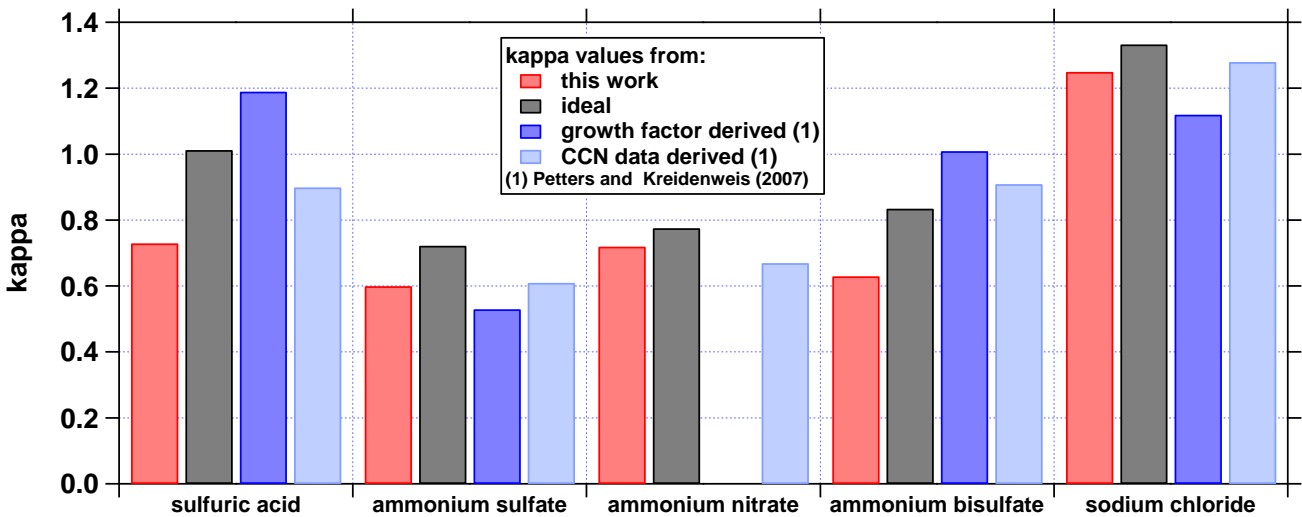

**Fig. 2**: Comparison of average hygroscopicity parameters ($\kappa_{mean}$) provided in Table 1 in Petters and Kreidenweis (2007) with the $\kappa$ values derived in this work based on a water activity of 0.9975 at the point of CCN activation as input to the E-AIM model II and IV (http://www.aim.env.uea.ac.uk/aim/model2/model2a.php). The water activity was derived from the following assumptions: $\kappa = 0.3$, SS = 0.5 %, $T = 5$ °C, and $\sigma = 74.95$ mN m$^{-1}$. The ideal $\kappa$ values refer to a water activity of 1. Note that the growth factor derived values in Petters and Kreidenweis (2007) are based on a water activity of about 0.9. For NaCl the value reported in Petters and Kreidenweis (2007) is too low and should be around 1.5 instead (Zieger et al., 2017).

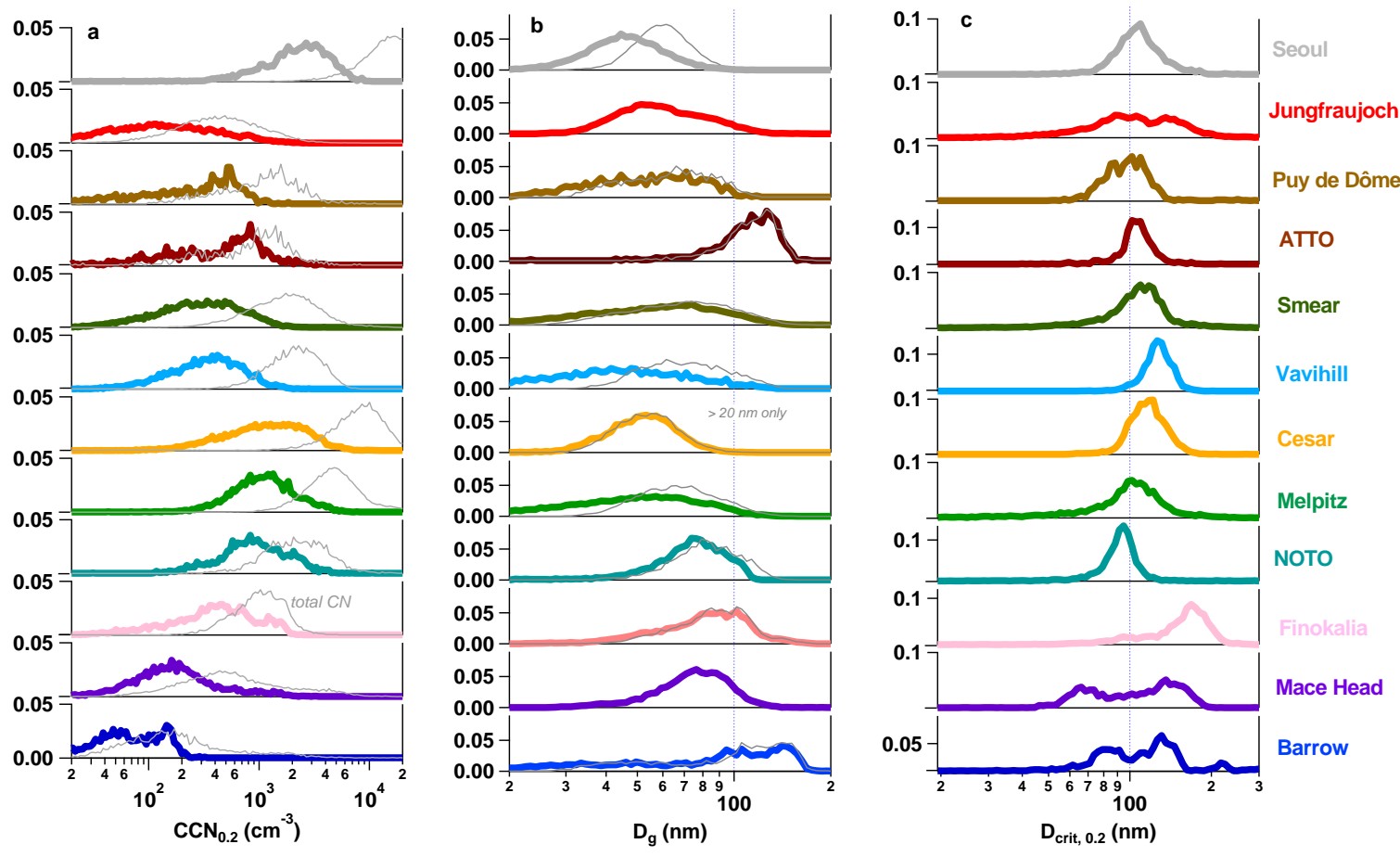

**Fig. 3:** (a) Normalized frequency distributions of CCN number concentration at SS = 0.2 % and total particle number in light grey, (b) geometric mean diameter $d_g$, and (c) critical diameter $d_{crit}$ at SS = 0.2 %. The grey lines in (b) are based on size distributions starting at 20 nm. The critical diameter is derived from the total CCN concentration (SS = 0.2 %) and the integrated particle number concentration starting from the largest diameter (see Section 2.2.2 for details). Note that seasons are not represented by an equal number of data points at each station which can lead to small biases in the frequency distributions. In (a) and (c) all axes start at 0.00.

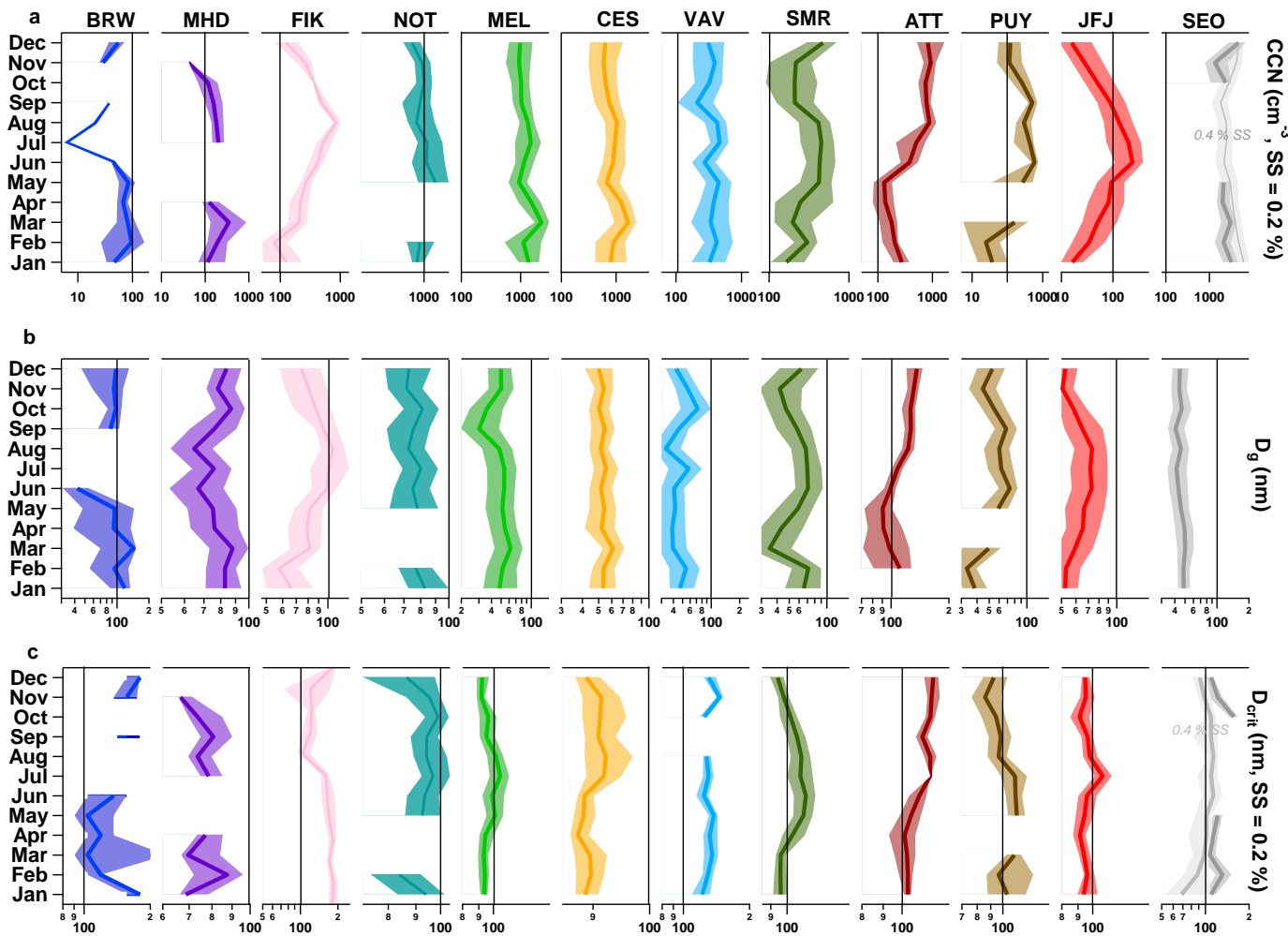

**Fig. 4**: Seasonal cycles (median and interquartile range) of **(a)** $CCN_{0.2}$ number concentration, **(b)** $D_g$, and **(c)** $D_{crit}$ at SS = 0.2 %. Note that only particles sizes > 20 nm were taken into account. The black vertical bars are placed at the same *x*-axis value in each panel for each station for better comparability. For SEO data at SS = 0.2 % was limited. In order to display the full seasonal cycle, values for SS = 0.4 %) are also shown. Note that the number of overlapping data points at VAV for CCN number concentration and particle number size distribution in October is < 200, i.e. < 10 days. No monthly median was derived. Also note, if the interquartile range seems to be missing, variations are so small that they do not appear beyond the thick median line.

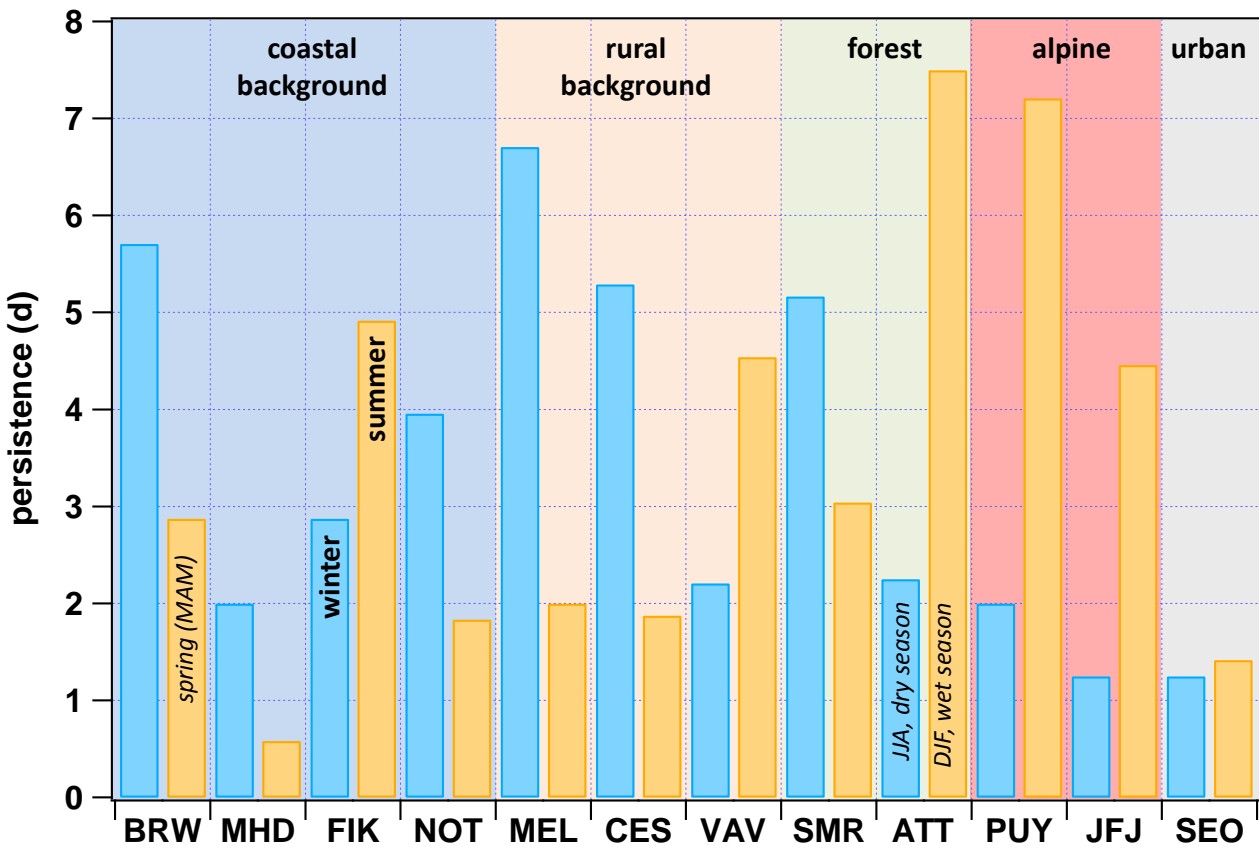

**Fig. 5**: Persistence of CCN number concentrations at SS = 0.2 % in days for winter (DJF) and summer (JJA). Note for BRW there were not sufficient data during summer, so spring values are shown, and since ATTO is located in the tropics, wet and dry seasons are different as indicated.

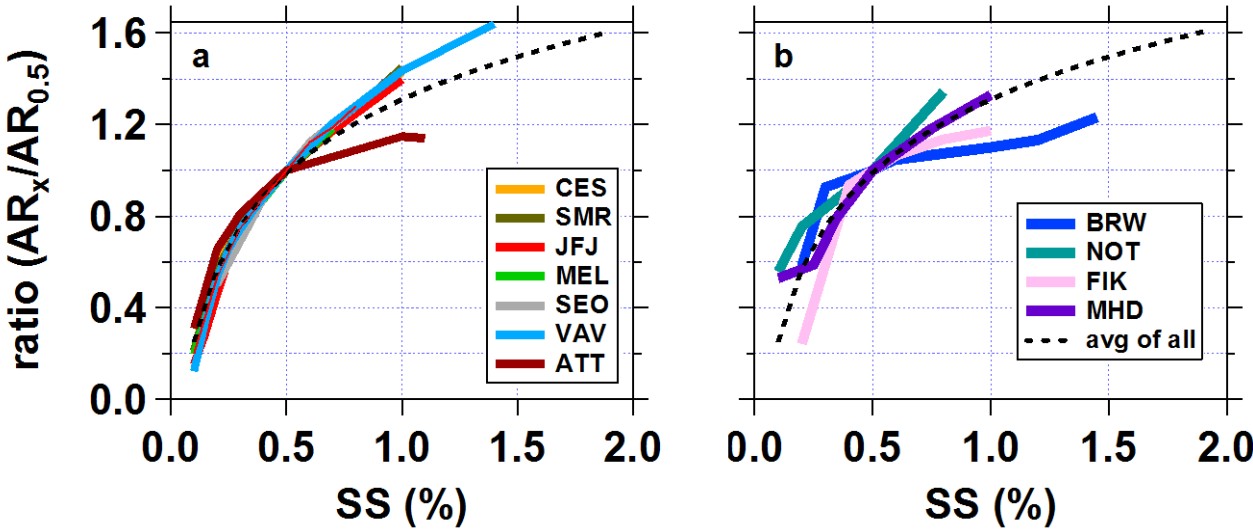

**Fig. 6**: Ratio of activation ratios for all measured SS % over the activation ratio at 0.5 % SS for each station. At SS = 0.5 % (x-axis) the ratio is 1. Activation ratios are based on particle size distributions starting at 20 nm. **(a)** shows non-coastal sites, while **(b)** groups all coastal sites. The black dotted line is the average fit through all curves from panels **(a)** and **(b)**, whereby y = A*ln(SS%)+b with A = 0.46 ± 0.02 and b = 1.31 ± 0.02.

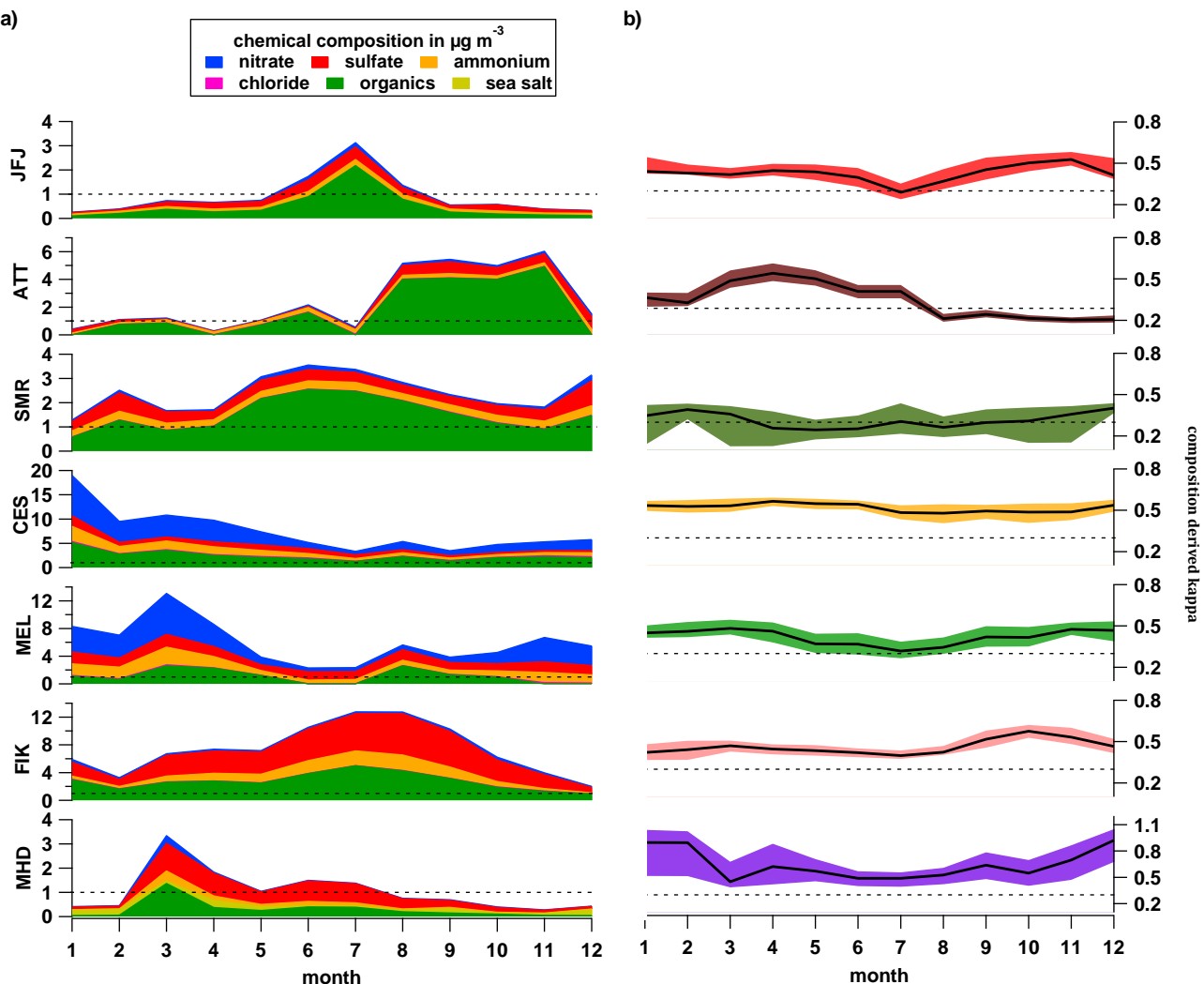

**Fig. 7: (a)** Monthly median chemical composition as measured by each station's mass spectrometer (see Table 1 for details on the type of spectrometer). The horizontal dashed line is placed at 1 µg m$^{-3}$ for easy comparison of mass concentrations between stations. **(b)** Median (black line) and interquartile range of composition-derived κ values per month. The dashed black line is located at κ = 0.3. Note, we do not show monthly BC concentrations where available here, because the displayed κ values are based only on the mass spectrometric data.

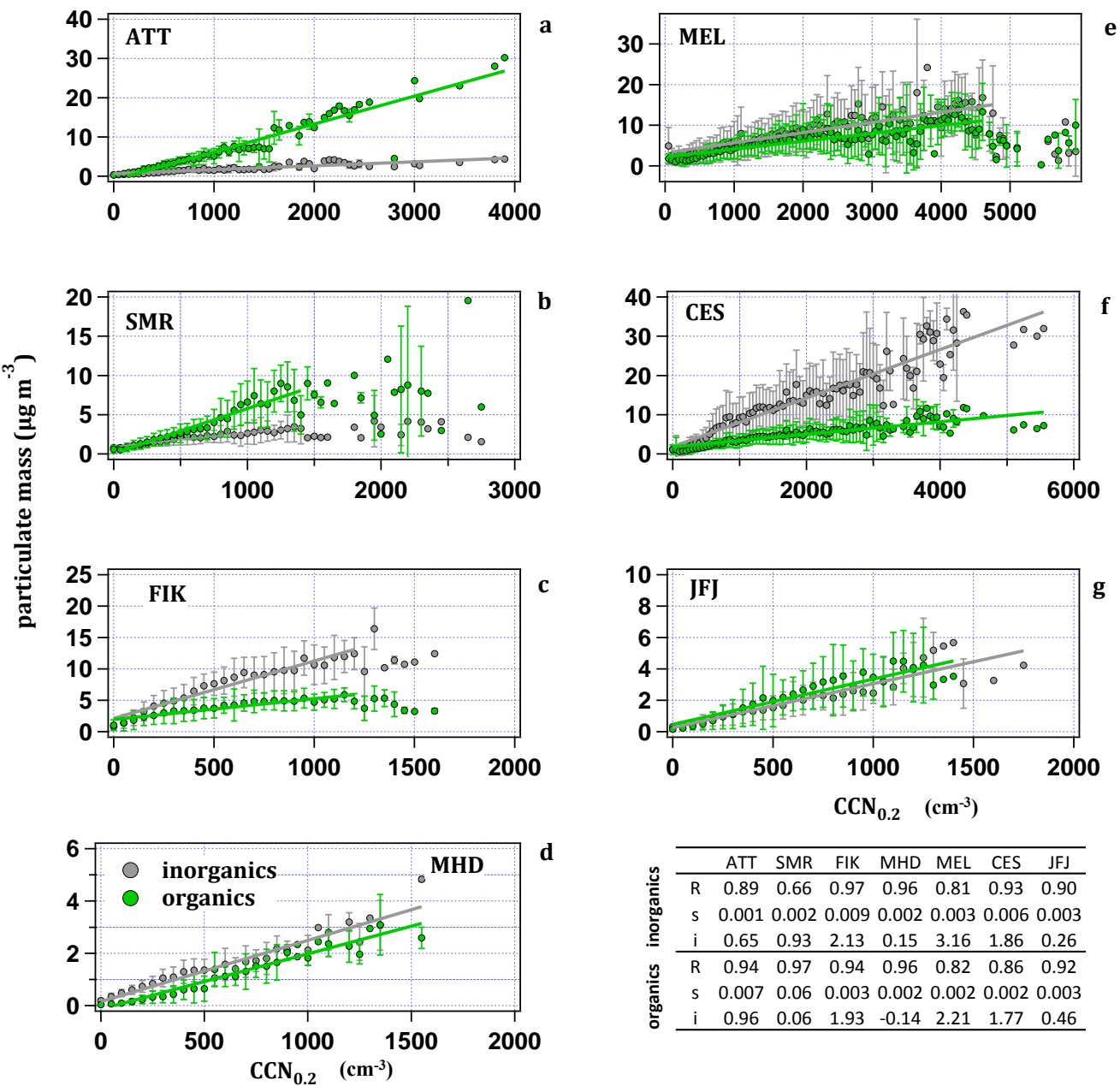

**Fig. 8**: Binned averages and standard deviations of inorganic and organic particle mass concentrations versus $CCN_{0.2}$. The mass concentrations are averaged over bins of 50 particles ($cm^{-3}$). Green and grey lines are linear fits through the points with the all parameters given in each panel. The table provides the linear regression data: $R$ stands for correlation coefficient, $s$ for slope, $i$ for intercept.

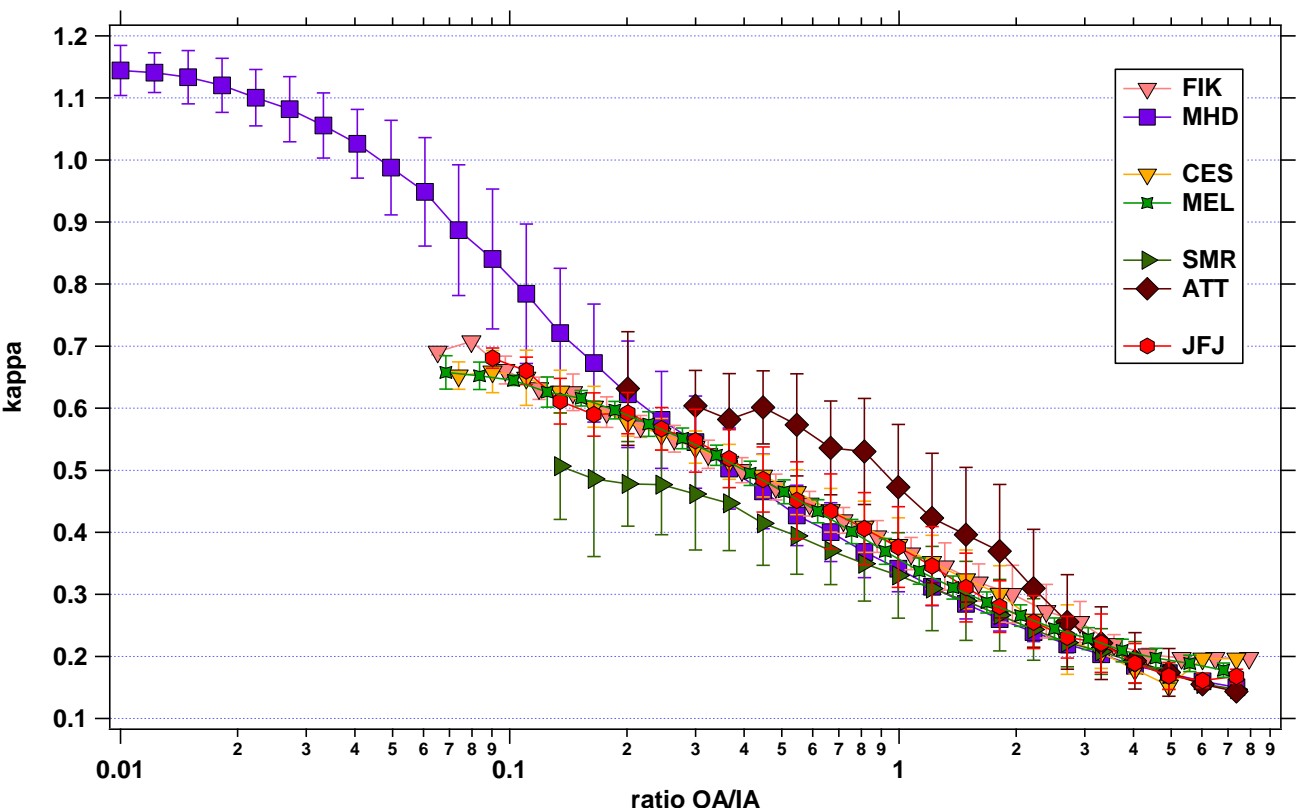

**Fig. 9**: Relationship of the composition-derived hygroscopicity parameter, $\kappa$, to the binned and averaged ratio of organic (OA) to inorganic (IA) aerosol components. The vertical bars denote the standard deviation. Note that the asymptotic-like approach of the curves towards a $\kappa$ value higher than 0.1 cannot be interpreted as $\kappa$ being larger than 0.1 for these sites, because $\kappa = 0.1$ was used as assumption to derive the $\kappa$ values shown on the $y$-axis. Note that the standard deviation for the lowest OA/IA ratios at FIK are so small that they do not go beyond the symbol.

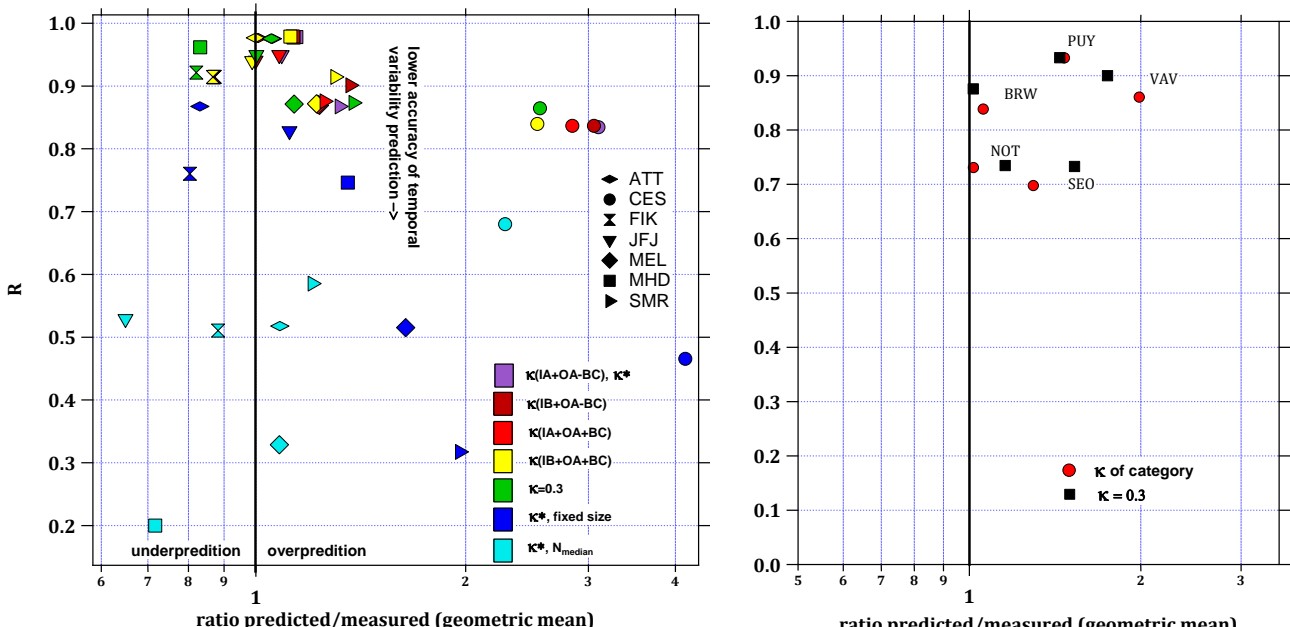

**Fig. 10: (a)** Results from closure studies for the seven stations with aerosol chemical composition data. The coefficient of the correlation between predicted to measured CCN number concentration at SS = 0.5 % is shown in the vertical axis while the geometric mean of the ratio of predicted and measured CCN number concentration is given on the horizontal axis. The different marker symbols represent the stations while the colors indicate details of the closure study. Kappa values refer to how the hygroscopicity parameter was calculated as described in Section 2.3.2 and as listed in Table 2. "fixed size" refers to closure studies where the fixed average shape of all size distributions from the data set was applied while keeping the temporally variable particle number concentrations as measured at each station. "$N_{median}$" means that closure studies were performed fixing the particle number concentration at each station to its median value while keeping the temporally variable shape of the size distribution. **(b)** Closure results for all stations without chemical composition data using kappa = 0.3 and an average kappa per site category (VAV: rural background, $\kappa = 0.48$; PUY: alpine, $\kappa = 0.41$ (e.g., JFJ); BRW and NOT: coastal background, $\kappa = 0.55$; SEO: urban, $\kappa = 0.1$).

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
