# Peer review of "Long-term cloud condensation nuclei number concentration, particle number size distribution, and chemical composition measurements at regionally representative observatories"

_Atmospheric Chemistry and Physics, 2017_

## Referee Comment (RC1) · Anonymous Referee #1 · 12 Nov 2017

Schmale et al. present in their manuscript an analysis of cloud condensation nuclei (CCN), particle size distribution and particle chemical composition measurements recorded at twelve sites from three continents. A closure study based on the measured particle size and chemical composition and applying the $\kappa$-Koehler parametrization is performed to test several simplified assumptions potentially be used to predict CCN concentrations. The applied methods and analysis steps appear as sound and valid. The results are described and discussed in sufficient detail. Overall, this work is of clear

interest to the scientific community. Only a few clarifications, as well as minor changes to text and figures are needed before it can be published in ACP (minor revisions; see comments below).

**Detailed comments**

1. The manuscript is in some parts quite lengthy and makes the reading for some parts slightly difficult. The abstract almost reads like an introduction and could be substantially shortened to the main findings. Most of the data treatment (incl. technical tables) and site descriptions are already published in the data descriptor paper and could thus be shortened as well.

2. Page 5, line 18: The particle's hygroscopicity is another important parameter needed for the simulation of CCN concentration, which should be mentioned here (incl. references).

3. Page 8, line 16: What errors can be expected when integrating the monodisperse CCN measurements over the covered size range in order to be comparable to the polydisperse mode? I guess this will depend on the specific selected diameters and the number of selected diameters (resolution).

4. First paragraph on page 12: Sodium chloride is not part of the E-AIM II model. How was this treated?

5. Figure 2: It should be clarified (e.g. in the caption) that the values from Petters and Kreidenweis (2007) in the subsaturated domain were determined at $a_w \sim 0.9$. It should also be mentioned that the values in Petters and Kreidenweis (2007) for NaCl are too low and should rather be $\kappa \sim 1.5$ (at $a_w \sim 90\,\%$) as recently shown by Zieger et al. (2017).

6. Sect. 2.3.2: The influence of sea salt, which is not properly detected by the AMS or ACSM, on the overall $\kappa$ should be further discussed. Sea salt is not just limited to coarse mode particles (see e.g., Salter et al., 2015) and is clearly observed in measurements even at rural background sites like Melpitz or Cabauw (see e.g., Zieger et al., 2011, 2014) also for longer time periods.

7. Page 15, line 2-3: Where the air masses from the polluted sector at MHD excluded from the analysis? I wonder because MHD is classified here as 'coastal background' station.

8. Page 17, second paragraph and Fig. 5: The calculation of the persistence needs more detail on the calculation method and its uncertainty. In the beginning of the paragraph, it states that 'The persistence is essentially a metric for how long the CCN number concentration remains similar.'. However, this is in contradiction to the discussion of the mountain stations later on ('... the high persistence is an indication of a regular pattern rather than a constant CCN number concentration'). Maybe rephrase this part. Why are the other seasons not included in Fig. 5?

9. Page 20, line 16: Please specify what 'A certain relationship' means.

10. Page 22, line 25: I can't find the here referred white symbols in Fig. 10a.

11. Figure 4: Please clarify on the following:

    - BRW shows no interquartile ranges in the summer with respect to CCN concentration.
    - MHD misses the CCN concentration for the summer, but shows the derived critical diameter (panel c) in summer.
    - VAV misses the values for the critical diameter in fall but shows values for CCN concentration during that period.

12. Figure 7: The measured or assumed BC used for calculating $\kappa$ is missing in panel (a).

13. Figure S3: Please clarify if values in panel (d) are shown as mean values for the entire supersaturation range.

14. Table 1 is not really needed and partially repetitive to the information shown in the data descriptor paper. It could therefore be moved to the supplement. It also shows the collection efficiency for the various ACSM and AMS measurements, but the importance of these values are not discussed in the manuscript. Please add a discussion on it or remove this information from the table.

15. Table 2: The last column is not really needed here and could be removed.

**Technical comments**

1. Page 7, line 24: Here and throughout the manuscript, 'Fig.' should be 'Figure' at the beginning of the sentence (see https://www.atmospheric-chemistry-and-physics.net/for_authors/manuscript_preparation.html).

2. Page 7, line 31: This sentence is a repetition from line 13 on the same page.

3. Page 8, line 3: CCN was already defined in the abstract and the introduction.

4. Page 8, line 5: Repetition (see page 7, line 29).

5. Equation 1: '$(r_k)$' should not be in subscript.

6. Page 12, line 17: Space missing.

7. Page 13, line 6: Replace 'at' with 'as'.

8. Page 13, line 25: I assume you mean Eq. 6 in Petters and Kreidenweis (2007)?

9. Page 19, line 17 and throughout the manuscript: Since the subscripts relate to chemical components and not to variables, add '\rm' in the subscript if using Latex or remove the italics.

10. Figure 1: The colour code in panel (a) is difficult to distinguish. I would suggest to use different symbols or write the station names next to their symbol on the map.

11. Figure 6 and throughout the manuscript: Please be consistent in placing the units in parenthesis.

12. Figure 8: Units are missing for the x-axis.

13. Figure 9: The colours are difficult to distinguish, please use different markers. Why are the first values for FIK shown without errorbars?

14. Figure 10: Change the x-axis label to 'ratio predicted/measured (geometric mean)', since a ratio and not a geometric mean is shown (otherwise a unit would be missing).

15. In the tables, please harmonize the case sensitivity and the usage of hyphen.

16. Table 3: Please replace 'and' by 'to' and/or add '$CCN_{0.5}^{pred.}/CCN_{0.5}^{meas.}$' to the table caption.

**References**

Petters M. and Kreidenweis S., A single parameter representation of hygroscopic growth and cloud condensation nucleus activity, *Atmos. Chem. Phys.*, 7(8), 1961–1971, doi: 10.5194/acp-7-1961-2007, 2007.

Salter M.E., Zieger P., Acosta Navarro J.C., Grythe H., Kirkevåg A., Rosati B., Riipinen I., and Nilsson E.D., An empirically derived inorganic sea spray source function incorporating sea surface temperature, *Atmos. Chem. Phys.*, 15(19), 11047–11066, doi:10.5194/acp-15-11047-2015, 2015.

Zieger P., Fierz-Schmidhauser R., Poulain L., Müller T., Birmili W., Spindler G., Wiedensohler A., Baltensperger U., and Weingartner E., Influence of water uptake on the aerosol particle light scattering coeffcients of the Central European aerosol, *Tellus B*, 66, 22716, doi:10.3402/tellusb.v66.22716, 2014.

Zieger P., Väisänen O., Corbin J., Partridge D.G., Bastelberger S., Mousavi-Fard M., Rosati B., Gysel M., Krieger U., Leck C., Nenes A., Riipinen I., Virtanen A., and Salter M., Revising the hygroscopicity of inorganic sea salt particles, *Nature Communications*, 8(15883), doi:10.1038/ncomms15883, 2017.

Zieger P., Weingartner E., Henzing J., Moerman M., de Leeuw G., Mikkilä J., Ehn M., Petäjä T., Clémer K., van Roozendael M., Yilmaz S., Frieß U., Irie H., Wagner T., Shaiganfar R., Beirle S., Apituley A., Wilson K., and Baltensperger U., Comparison of ambient aerosol extinction coefficients obtained from in-situ, MAX-DOAS and LIDAR measurements at Cabauw, *Atmos. Chem. Phys.*, 11(6), 2603–2624, doi:10.5194/acp-11-2603-2011, 2011.

---

## Referee Comment (RC2) · Anonymous Referee #2 · 14 Nov 2017

The paper by Schmale et al. presents analyses based on a huge data set collected at twelve stations, eight of which belong to the ACTRIS network. It contains valuable data and data analyses and should certainly be published, but not without extensive revisions.

The abstract is quite long and reads somewhat like an introduction, and in the paper itself there are several repetitions and some rather lengthy passages. Another round of

rigorous editing is certainly called for. The paper would also benefit from a structuring process, where info that is currently distributed throughout the text is collected and presented in a structured form. One example: the importance of CCN for predicting CDNC is discussed on p. 26 (!) instead of the introduction – the whole point in CCN measurements from a climate perspective is their influence on cloud properties, so this should be discussed also in the introduction

Major points:

1. There is an abundance of qualitative statements that should be substantiated. Concentrations are describes as high / low / higher at ... than at ....,; correlations are described as "high", "good", etc., but no numbers are given. Data from different stations are compared and similarities and differences are described, but again only qualitatively. Mean values are compared without giving standard deviations, etc. Most quantitative information is contained in figures and tables, but readers should not have to go back and forth in search of important information or estimate values from figures.

2. In many cases, CCN concentrations etc. are given without mentioning the corresponding SS, which is necessary to put the data in context. The paper should be edited also regarding these omissions.

3. Instrument description is practically non-existent. Of course there is a companion paper giving the experimental details, but at least the most crucial limitations of the instruments should be given also in this MS to enable readers to judge the validity of results without having to consult another paper. The lower cut size, e.g., of the mass spectrometers must be given in order to correctly interpret the section on the calculation of kappa from the chemical composition. Most CCN active at the higher supersaturations used in the DMT CCNC-100 will have sizes way below the lower cut size of the aerodynamic lenses used in some mass spectrometers (around 1 $\mu$m).

4. Problems with instruments should not be mentioned in half sentences but should be properly discussed. On p. 22, lines 9-10, e.g., the over-prediction of CCN using kappa

is attributed to losses of small particles in the aerosol sampled by the CCNC – what is the basis for this statement? If there really were losses – can they be quanitifed? What would be the impact on all data measured at the CES observatory? Please add at least some info on intrumental problems and their effects also to this paper

5. Disregarding surface tension in the calculation of d_crit could be problematic. The paper states that haze particles at activation probably have the same surface tension as water, which is not correct (see e.g. Capel et al., 1990, Facchini et al., 2000 and Hitzenberger et al., 2002) and the effect of surface tension on CCN activation, which was indicated e.g. by Charlson et al, 2001, is inadequately discussed.

6. The title of the paper is misleading, as it suggests a far reaching review of what "we" (the scientific community?) have learned about CCN and CCN closure. This is not the case – the paper presents valuable data and valuable data analysis, but it is nevertheless limited to the ACTRIS network plus one station each in Korea and Japan, and two stations in the US. Global coverage is patchy, and as four of the stations are coastal background, three are rural background, two are high alpine, two are remote, and one is urban, the question of how representative these stations are for a global assessment remains open. Please change the title to avoid misunderstandings

7. Earlier work is referenced somewhat selectively. The paper mentions and references some previously published data sets, too, but implies that most studies were based on short-term intensive field campaigns, or that there are no (short or long-term) data sets for urban areas. This is not correct – see the studies by Che et al. (2016) and (2017) and Deng et al. (2013) conducted in more or less polluted regions in China (including also parametrizations of CCN activation), the studies at urban sites published by Burkart et al. (2011) and (2012) for the urban aerosol in Vienna, Austria (the latter study also includes kappa from chemical analyses of particles < 100 nm), and by Leng et al. (2013) in downtown Shanghai; the two year study by Fors et al. (2011) in rural Sweden and for the boreal forest by Sihto et al. (2011). This is just a short list of pertinent studies not referenced in the paper. At least those that include discussion of

CCN parameters other than concentrations should be included in the discussion.

There are several minor (technical) things such as missing words or cut-and-paste relics, but a thorough editing of the whole paper will reveal and remove them anyway.

References:

Burkart, J., Steiner, G., Reischl, G., Hitzenberger, R., 2011. Long-term study of cloud condensation nuclei (CCN) activation of the atmospheric aerosol in Vienna. Atmos. Environ. 45, 5751–5759. doi:10.1016/j.atmosenv.2011.07.022

Burkart, J., Hitzenberger, R., Reischl, G., Bauer, H., Leder, K., Puxbaum, H., 2012. Activation of "synthetic ambient" aerosols - Relation to chemical composition of particles <100nm. Atmos. Environ. 54, 583–591. doi:10.1016/j.atmosenv.2012.01.063

Capel, P. D., R. Gunde, F. ZuÂÍrcher, and W. Giger, Carbon speciation and surface tension of fog, 1990. Environ. Sci. Technol., 24, 722–727

Charlson, Robert J., John H. Seinfeld, Athanasios Nenes, Markku Kulmala, Ari Laaksonen, M. Cristina Facchini , 2001. Reshaping the Theory of Cloud Formation. Science, 292, Issue 5524, 2025-2026

Che, H. C., X. Y. Zhang, L. Zhang, Y. Q. Wang, Y. M. Zhang, X. J. Shen, Q. L. Ma, J. Y. Sun & J. T. Zhong, 2017. Prediction of size-resolved number concentration of cloud condensation nuclei and long-term measurements of their activation characteristics. Scientific Reports 7, Article number: 5819 doi:10.1038/s41598-017-05998-3

Che, H. C. X. Y. Zhang, Y. Q. Wang, L. Zhang, X. J. Shen, Y. M. Zhang, Q. L. Ma, J. Y. Sun, Y. W. Zhang & T. T. Wang, 2016. Characterization and parameterization of aerosol cloud condensation nuclei activation under different pollution conditions. Scientific Reports 6, Article number: 24497 (2016) doi:10.1038/srep24497

Deng, Z. Z., C. S. Zhao, N. Ma, L. Ran, G. Q. Zhou, D. R. Lu, and X. J. Zhou, 2013. An examination of parameterizations for the CCN number concentration based on in

situ measurements of aerosol activation properties in the North China Plain. Atmos. Chem. Phys., 13, 6227-6237, https://doi.org/10.5194/acp-13-6227-2013

Facchini, M. C., S. Decesari, M. Mircea, S. Fuzzi, and G. Loglio, 2002. Surface tension of atmospheric wet aerosol and cloud/fog droplets in relation to their organic carbon content and chemical composition, Atmos. Environ., 34, 4853–4857.

Fors, E. O., E. Swietlicki, B. Svenningsson, A. Kristensson, G. P. Frank, and M. Sporre, 2011. Hygroscopic properties of the ambient aerosol in southern Sweden – a two year study. Atmos. Chem. Phys., 11, 8343-8361 https://doi.org/10.5194/acp-11-8343-2011

Hitzenberger, R., Berner, A., Kasper-Giebl, A., Loflund, M., Puxbaum, H., 2002. Surface tension of Mt. Rax cloud water and its relation to the concentration of inorganic and organic material. J. Geophys. Res D24, 2002JD002506, paper ID 4752

Leng, C., Cheng, T., Chen, J., Zhang, R., Tao, J., Huang, G., Zha, S., Zhang, M., Fang, W., Li, X., Li, L., 2013. Measurements of surface cloud condensation nuclei and aerosol activity in downtown Shanghai. Atmos. Environ. 69, 354–361. doi:10.1016/j.atmosenv.2012.12.021

Sihto, S.-L., J. Mikkilä, J. Vanhanen, M. Ehn, L. Liao, K. Lehtipalo, P. P. Aalto, J. Duplissy, T. Petäjä,V.-M. Kerminen, M. Boy, M. Kulmala (2011), Seasonal variation of CCN concentrations and aerosol activation properties in boreal forest Atmos. Chem. Phys. 11, 13269–13285.

---

## Author Comment (AC1) · 11 Dec 2017

Please download the supplement which contains the following documents:

1) Responses to both reviewers

2) The revised manuscript with tracked changes

3) The revised manuscript without tracked changes

[Figure]

Please also note the supplement to this comment:
https://www.atmos-chem-phys-discuss.net/acp-2017-798/acp-2017-798-AC1-
supplement.zip

---

## Author Response (AR1)

**Response to Reviews regarding MS No.: acp-2017-798**

"What do we learn from long-term cloud condensation nuclei number concentration, particle number size distribution, and chemical composition measurements at regionally representative observatories?"

Abbreviations:

RC1 1: Reviewer 1 comment number 1

AR1 1: Authors' response to RC1 1

MC1 1: Manuscript change based on AR1 1, page and line numbers in our answers correspond to the document with track changes.

**Reviewer 1:**

We thank reviewer 1 for the detailed and very constructive review. Please find our responses below.

RC1 1: The manuscript is in some parts quite lengthy and makes the reading for some parts slightly difficult. The abstract almost reads like an introduction and could be substantially shortened to the main findings. Most of the data treatment (incl. technical tables) and site descriptions are already published in the data descriptor paper and could thus be shortened as well.

AR1 1: We have shortened the abstract following the recommendations of both reviewers. We did think about the two options of (a) keeping the site, instrument and data treatment sections very short as this is published in the companion paper, and (b) repeating part of this information for readers not to have to consult another paper. The compromise as presented in this manuscript did receive two different comments. Reviewer 1 prefers to shorten this section while reviewer 2 (RC2 3) asks for more information. We have done the following to address the opposing comments: With regards to the descriptions of the instruments, measurements sites and data treatment, we have shortened the text slightly (see below for details), whereby some additional information on the mass spectrometers was requested by reviewer 2 (RC2 3) which we have inserted. Table 1 remains as it is and collection efficiencies are explained in the text (refers to RC1 14).

MC1 1:

**Abstract**

"Aerosol-cloud interactions (ACI) constitute the single largest uncertainty in anthropogenic radiative forcing. To reduce the uncertainties and gain more confidence in the simulation of ACI, models need to be evaluated against observations, in particular against measurements of cloud condensation nuclei (CCN). Here we present a data set - ready to be used for model validation - of long-term observations of CCN number concentrations, particle number size distributions and chemical composition from twelve sites on three continents. Studied environments include: coastal background, rural background, alpine sites, remote forests and an urban surrounding. Expectedly, CCN characteristics are highly variable across site categories. However, they also vary within them, most strongly in the coastal background group, where CCN number concentrations can vary by up to a factor of 30 within one season. In terms of particle activation behavior, most continental stations exhibit very similar activation ratios (relative to particles > 20 nm) the range of 0.1 to 1.0 %

supersaturation. At the coastal sites the transition from particles being CCN inactive to becoming CCN active occurs over a wider range of the supersaturation spectrum.

Several stations show strong seasonal cycles of CCN number concentrations and particle number size distributions, e.g., at Barrow (Arctic Haze in spring), at the alpine stations (stronger influence of polluted boundary layer air masses in summer), the rain forest (wet and dry season), or Finokalia (forest fire influence in fall). The rural background and urban sites exhibit relatively little variability throughout the year while short-term variability can be high especially at the urban site.

The average hygroscopicity parameter,  $\kappa$ , calculated from the chemical composition of submicron particles, was highest at the coastal site of Mace Head (0.6) and the lowest at the rain forest station ATTO (0.2 - 0.3). We performed closure studies based on the  $\kappa$ -Köhler theory to predict CCN number concentrations. The ratio between predicted and measured CCN concentrations is between 0.87 and 1.4 for five different types of  $\kappa$ . The temporal variability is also well captured, as reflected by Pearson correlation coefficients > 0.87.

Information on CCN number concentrations at many locations is important to better characterize ACI and their radiative forcing. But long-term comprehensive aerosol particle characterizations are labor intensive and costly. Hence, we recommend operating "migrating-CCNCs" to conduct collocated CCN number concentration and particle number size distribution measurements at individual locations throughout one year at least to derive a seasonally resolved hygroscopicity parameter. This way, CCN number concentrations can be calculated based on continued particle number size distribution information only and greater spatial coverage of long-term measurements can be achieved."

Text:

p.8 l. 25: Removed the reference to Schmale et al. (2017).

P. 8, I.33: included "in the submicron size range."

p.9, l. 2ff: included "Table 1 also lists the collection efficiency (CE) of each mass spectrometer. The CE depends on the transmission of particles into the instrument and their chemical composition and is hence an instrument and site specific factor (Huffman et al., 2005; Middlebrook et al., 2012)."

p. 9, l. 11 - 23: we have shortened this paragraph to "The collection, harmonization and quality assurance of the data sets presented here are described in detail in the data descriptor by Schmale et al. (2017). Data have a time resolution of one hour and represent standard temperature and pressure (STP) conditions. The time resolution of CCN number concentrations at Puy de Dôme (PUY) and ATTO are four and six hours, respectively, because the scans over the submicron aerosol size range in monodisperse mode took longer."

RC1 2: Page 5, line 18: The particle's hygroscopicity is another important parameter needed for the simulation of CCN concentration, which should be mentioned here (incl. references).

AR/MC1 2: We have included the following sentence (p.5, l. 20f): "Information on aerosol hygroscopicity is also needed to constrain uncertainty (Rosenfeld et al., 2014)."

RC1 3: Page 8, line 16: What errors can be expected when integrating the monodisperse CCN measurements over the covered size range in order to be comparable to the polydisperse mode? I

guess this will depend on the specific selected diameters and the number of selected diameters (resolution).

AR1 3: As the reviewer says, the error depends on the selected diameter resolution and the representativeness of the specifically selected diameters for the ambient particle number size distribution. To test how much influence these have, we used the JFJ SMPS data set and compared the integrated particle number concentrations for 26304 scans over 81 bins with the concentrations integrated over a reduced number of bins. Using only 27 bins (a third) the discrepancy between the integrated number concentrations is only 2 % (overestimation). Using only 14 bins, the overestimation is 6 %. The reduction of bins in the second case is exaggerated, but the result is still within the 10 % expected uncertainty from any comparison between particle counting systems we consider in our study (Schmale et al., 2017; Wiedensohler et al., 2012).

In addition, the ambient variability of the particle properties plays a role. If the size distribution or hygroscopicity were to change on the order of 5 minutes, for example, and a scan of monodisperse CCN measurements took 30 minutes, this method would average over 6 different particle populations while the polydisperse method recording at a higher time resolution would capture the variability. In general, such variations would affect the random noise level but not the average for these data sets, because they are sufficiently large. However, in the case of PUY and ATTO scans through all supersaturations took 4 and 6 hour, respectively. So in order to calculate the activation ratio or the critical diameter, we used only the size distributions recorded while the CCNC was scanning at the specific supersaturation of interest. The result of this shorter interval is then taken as representative of the 4 or 6 hours total scanning interval. The same procedure was used for MEL and NOT, even though the CCNCs scanned faster (1 hour averages).

Based on the companion paper (Schmale et al., 2017) we do not see that monodisperse CCN number concentration determination introduces larger errors than determination through polydisperse measurements. In Fig. 4 in the companion paper we assess the reliance of the CCN number concentration measurement for polydisperse measurements based on the ratio of CCN1.0 and the total particle number (as derived from each station's SMPS or DMPS across the whole size range) when the contribution of particles < 30 nm is smaller than 10 %. In Fig. 5 we assess the reliance of monodisperse measurements by comparing the integrated CCN number concentration at a certain supersaturation to the integrated particle number concentration as determined by the SMPS. The lower diameter is chosen such that it is larger than the estimated activation diameter. Comparing these two figures, there is no evidence that monodisperse quantification leads to larger discrepancies than polydisperse quantification.

We conclude that there is no significant bias between the poly- and monodisperse data sets used in this study.

RC1 4: First paragraph on page 12: Sodium chloride is not part of the E-AIM II model. How was this treated?

AR1 4: Thank you for pointing this out. We used Model IV for sodium and chloride. We changed the text as follows:

MC1 4, p. 12, l. 13ff: "This reference water activity was used as input to the E-AIM model II and IV (http://www.aim.env.uea.ac.uk/aim/aim.php), by which the particulate water content was calculated for the pure salts and inorganic acids in aqueous solution. The E-AIM II model is an equilibrium thermodynamic model including the following ions:  $H^+$ ,  $NH_4^+$ ,  $SO_4^{2-}$ ,  $NO_3^-$ ,  $H_2O$ . It is valid from 328 K to about 200 K. Model IV includes  $Na^+$  and  $Cl^-$  and is valid from 180 - 330 K."

RC1 5: Figure 2: It should be clarified (e.g. in the caption) that the values from Petters and Kreidenweis (2007) in the subsaturated domain were determined at aw 0.9. It should also be mentioned that the values in Petters and Kreidenweis (2007) for NaCl are too low and should rather be kappa 1.5 (at aw 90%) as recently shown by Zieger et al. (2017).

AR1 5: Thank you for pointing this out. We have added an explanation in the caption and in the corresponding section of the manuscript.

**MC1 5:**

Caption Figure 2: "Note that the growth factor derived values in Petters and Kreidenweis (2007) are based on a water activity of about 0.9. For NaCl the value is too low and should be around 1.5 (Zieger et al., 2017)."

P.12, I. 22f: "Note that the value for NaCl in Petters and Kreidenweis is too low, instead of 1.12 it should be around 1.5 (Zieger et al., 2017)."

RC1 6: Sect. 2.3.2: The influence of sea salt, which is not properly detected by the AMS or ACSM, on the overall kappa should be further discussed. Sea salt is not just limited to coarse mode particles (see e.g., Salter et al., 2015) and is clearly observed in measurements even at rural background sites like Melpitz or Cabauw (see e.g., Zieger et al., 2011, 2014) also for longer time periods.

AR1 6: We did take into account exactly the same considerations that the reviewer mentions, however, we did not explicitly mention this. We now added the following sentences:

**MC1 6:**

p. 12, l. 24: "In our study, we assume that chloride is present in the form of NaCl and apply the  $\kappa$  value as shown in Fig. 2. For MHD, the contribution of submicron sea salt has been calculated by the data originators after Ovadnevaite et al. (2011) to which we assign the same  $\kappa$  value. Given that the AMS and ACSM do not fully detect sea salt components which are present in the submicron aerosol (Salter et al., 2015), this contribution is likely to be underestimated at all other stations close to the sea and where chemical composition data are available (e.g., CES, FIK)."

RC1 7: Page 15, line 2-3: Where the air masses from the polluted sector at MHD excluded from the analysis? I wonder because MHD is classified here as 'coastal background' station.

AR1 7: We adopted the official WMO station classification. Polluted sectors are included in the data set because the point of this study is to elucidate the varying conditions at each site and how this is reflected in the aerosol variables. For example on p. 15, l. 16 we state that MHD represents only for certain periods clean coastal conditions, and on p. 16, l. 18 we discuss anthropogenic contributions to aerosol present at MHD.

RC1 8: Page 17, second paragraph and Fig. 5: The calculation of the persistence needs more detail on the calculation method and its uncertainty. In the beginning of the paragraph, it states that 'The persistence is essentially a metric for how long the CCN number concentration remains similar.'.

However, this is in contradiction to the discussion of the mountain stations later on ('... the high persistence is an indication of a regular pattern rather than a constant CCN number concentration'). Maybe rephrase this part. Why are the other seasons not included in Fig. 5?

AR1 8: We noticed that we did not explicitly say how we derived the persistence from the autocorrelation curve. A sentence in the methodology section has been added that explains that the time coordinate is equal to the persistence where the auto-correlation curve crosses the large lag standard error. We have rephrased the sentence as follows below. The point we wanted to make is that patterns rather than absolute concentrations are reflected in the persistence calculation. In section 2.3.1, we elaborate how we calculated the persistence and how we treat its uncertainty. We prefer keeping these methodological details in the methods section rather than repeating or moving them to the results section.

Other seasons are not included because incidentally gaps in the time series occurred predominantly in the transition seasons. Furthermore, results do not provide insights that would add information to what we present for the summer and winter seasons. At all stations the winter and summer seasons are characteristically different and the persistence can elucidate synoptic influences.

MC1 8, p. 11, l. 19: "The persistence of a property is determined by the time coordinate at which the auto-correlation curve crosses the large lag standard error curve."

p. 18, l. 12: "The persistence is essentially a metric for how long the pattern of CCN number concentrations "remains similar" (see Sec 2.3.1). This does not exclude periodic variations on shorter time scales, such as diurnal cycles, than the observed persistence as long as the amplitude of the periodic variations and the averages over the cycles remain similar."

RC1 9: Page 20, line 16: Please specify what 'A certain relationship' means.

AR/MC1 9, p. 21, l. 14: We have clarified the sentence that now reads: "A negative relationship..."

RC1 10: Page 22, line 25: I can't find the here referred white symbols in Fig. 10a.

AR1 10: The dark blue symbols in the figure where meant instead of "white". We have replaced the color in the manuscript text accordingly.

RC1 11: Figure 4: Please clarify on the following:

BRW shows no interquartile ranges in the summer with respect to CCN concentration.
 MHD misses the CCN concentration for the summer, but shows the derived critical diameter (panel c) in summer.

*3. VAV misses the values for the critical diameter in fall but shows values for CCN concentration during that period.*

AR1 11:

1. It does, but the range is so small that it is difficult to notice in the figure (this happens for individual months at other stations as well). We have added a sentence to the caption explaining this.

- 2. To obtain an idea of the full seasonal cycle we had calculated the MHD critical diameter based on the kappa Köhler theory for the missing months and mistakenly plotted these results instead of the results purely derived from the CCN and size distribution measurements. We have now eliminated the kappa-Köhler derived values from the plot.
- 3. The CCN number concentrations are partly available for the two October months in 2013 and 14. Also the size distribution measurements are partly available. There are however gaps and those occur in such a way that when CCN data is available the size information is lacking and vice versa, resulting in a very small number (~200 at a time resolution of 1 hour) of parallel data points, so that deriving a monthly median is not meaningful. We included this information now in the captions.

MC 1 11: Note that the number of overlapping data points at VAV for CCN number concentration and particle number size distribution in October is < 200, i.e. < 10 days. No monthly median was derived.

RC1 12: Figure 7: The measured or assumed BC used for calculating kappa is missing in panel (a).

AR1 12: The "standard" kappa ( $\kappa_{IA+OA-BC}$ , section 2.3.2) we are applying does not include BC. This is the kappa we show in panel (b). For that reason we only display the mass spectrometric data, because including BC would be confusing. We have added a note under the figure.

RC1 13: Figure S3: Please clarify if values in panel (d) are shown as mean values for the entire supersaturation range.

AR1 13: Thank you for pointing this out. Panel (d) addresses the changes for SS = 0.5 % because we have performed the closure study at that SS. We have added this information in the caption.

RC 1 14: Table 1 is not really needed and partially repetitive to the information shown in the data descriptor paper. It could therefore be moved to the supplement. It also shows the collection efficiency for the various ACSM and AMS measurements, but the importance of these values are not discussed in the manuscript. Please add a discussion on it or remove this information from the table.

AR1 14: While reviewer 1 recommends shortening the information on stations and instruments, reviewer 2 is asking for more details (see RC2 3). We prefer keeping the table in the main manuscript so readers have direct access to the instrumental details via the table, while we keep the descriptive text short. Again referring to RC2 3, we included two explanatory sentences on the collection efficiency, see answer MC1 1.

RC1 15. Table 2: The last column is not really needed here and could be removed.

AR1 15: The column has been removed.

Technical comments:

RC1 16. Page 7, line 24: Here and throughout the manuscript, 'Fig.' should be 'Figure' at the beginning of the sentence (see https://www.atmospheric-chemistry-and-physics.net/for\_authors/manuscript\_preparation. html). AR1 16: Done

*RC1 17: Page 7, line 31: This sentence is a repetition from line 13 on the same page.* AR1 17: The sentence has been removed.

*RC1 18: Page 8, line 3: CCN was already defined in the abstract and the introduction.* AR1 18: Now only the abbreviation is used.

*RC1 19: Page 8, line 5: Repetition (see page 7, line 29).* AR1 20: We have removed the repetitive part on p. 7, and keep in the information on p. 8.

*RC1 21: Equation 1: '(rk)' should not be in subscript.* AR1 21: Done

*RC1 22: Page 12, line 17: Space missing.* AR1 22: Inserted.

*RC1 23: Page 13, line 6: Replace 'at' with 'as'.* AR1 23: Done

RC1 24: Page 13, line 25: I assume you mean Eq. 6 in Petters and Kreidenweis (2007)? AR1 24: Yes.

*RC1 25: Page 19, line 17 and throughout the manuscript: Since the subscripts relate to chemical components and not to variables, add '\rm' in the subscript if using Latex or remove the italics.* AR1 25: Done.

RC1 26: Figure 1: The colour code in panel (a) is difficult to distinguish. I would suggest to use different symbols or write the station names next to their symbol on the map. AR1 26: We have surrounded the symbols with a white line now which makes the colors easier to distinguish.

*RC1 27: Figure 6 and throughout the manuscript: Please be consistent in placing the units in parenthesis.* AR1 27: Done.

*RC1 28: Figure 8: Units are missing for the x-axis.* AR1 28: Units are now included.

*RC1 29: Figure 9: The colours are difficult to distinguish, please use different markers. Why are the first values for FIK shown without errorbars?*

AR1 29: We have introduced different symbols now. The standard deviation on the first FIK values are so small that they are within the symbol range. We have added a note to the figure caption.

RC1 30: Figure 10: Change the x-axis label to 'ratio predicted/measured (geometric mean)', since a ratio and not a geometric mean is shown (otherwise a unit would be missing). AR1 30: Done. *RC1 31: In the tables, please harmonize the case sensitivity and the usage of hyphen.* AR1 31: Done.

RC1 32 Table 3: Please replace 'and' by 'to' and/or add 'CCNpred. 0.5 /CCNmeas. 0.5 ' to the table caption. AR1 32: We replaced "and" by "to".

**Reviewer 2**

The paper by Schmale et al. presents analyses based on a huge data set collected at twelve stations, eight of which belong to the ACTRIS network. It contains valuable data and data analyses and should certainly be published, but not without extensive revisions.

We thank the reviewer for the comments and believe that the manuscript has now been improved significantly after addressing the points as follows.

RC2 1: The abstract is quite long and reads somewhat like an introduction, and in the paper itself there are several repetitions and some rather lengthy passages. Another round of rigorous editing is certainly called for. The paper would also benefit from a structuring process, where info that is currently distributed throughout the text is collected and presented in a structured form. One example: the importance of CCN for predicting CDNC is discussed on p. 26 (!) instead of the introduction – the whole point in CCN measurements from a climate perspective is their influence on cloud properties, so this should be discussed also in the introduction.

**AC2 1:**

Re Abstract: Reviewer 1 had the same comment. Please see MC1 1 for the revised abstract and shortened passages.

Re CDNC: We are not exactly sure how the comment is meant "the importance for predicting CDNC is discussed on p. 26 (!) instead of in the introduction", because the first sentence of the introductions states "Cloud droplets are formed by activation of a subset of aerosol particles called cloud condensation nuclei (CCN), which affect the radiative properties of clouds through modifying the cloud droplet number concentrations (CDNC), cloud droplet size...". The second paragraph of the introduction (I. 13) discusses the importance of CCN and updraft limited regimes for CDNC. We also start the introduction with discussing the climate relevance of cloud properties modulated by CCN and CDNC.

**Major points:**

RC2 2. There is an abundance of qualitative statements that should be substantiated. Concentrations are describes as high / low / higher at ... than at ....,; correlations are described as "high", "good", etc., but no numbers are given. Data from different stations are compared and similarities and differences are described, but again only qualitatively. Mean values are compared without giving standard deviations, etc. Most quantitative information is contained in figures and tables, but readers should not have to go back and forth in search of important information or estimate values from figures.

In many cases, CCN concentrations etc. are given without mentioning the corresponding SS, which is necessary to put the data in context. The paper should be edited also regarding these omissions.

AR2 2: We understand the reviewer's perspective and now included more precise numbers in the manuscript.

With regards to the CCN concentrations, we exclusively refer to SS = 0.5 % in the closure studies and all other analyses are based on SS = 0.2 %. This is mentioned in the text, because we are well aware of the fact that CCN number concentrations are not meaningful without stating the corresponding supersaturation. Nevertheless, we included this information now explicitly.

We refer the Reviewer to the tracked changes in the revised manuscript, because additions are so numerous that we cannot list them here.

RC2 3: Instrument description is practically non-existent. Of course there is a companion paper giving the experimental details, but at least the most crucial limitations of the instruments should be given also in this MS to enable readers to judge the validity of results without having to consult another paper. The lower cut size, e.g., of the mass spectrometers must be given in order to correctly interpret the section on the calculation of kappa from the chemical composition. Most CCN active at the higher supersaturations used in the DMT CCNC-100 will have sizes way below the lower cut size of the aerodynamic lenses used in some mass spectrometers (around 1  $\mu$ m).

AR2 3: In the response AR1 1, we elaborate that we did think about the option of either keeping the description very short because of the companion paper or of providing limited details. Reviewer 1 prefers to see even less detail than we provided, while reviewer 2 would like to have more details. We included the size range information of the aerosol mass spectrometers because we agree that this is important information. Generally, all AMS and ACSMs used in this study measure submicron particles (note the upper cut-off size is 1  $\mu$ m, not the lower). Please see answer AR1 1 for the exact changes in the manuscript.

Regarding the impact of the size range for the calculation of kappa, we have already discussed this in the manuscript on p. 13, l. 30- p. 14, l. 4. In short, since the mass spectrometer data are based on aerosol mass size distributions, the kappa is biased towards the larger particle size fraction that contributes most to the submicron mass.

RC2 4: Problems with instruments should not be mentioned in half sentences but should be properly discussed. On p. 22, lines 9-10, e.g., the over-prediction of CCN using kappa is attributed to losses of small particles in the aerosol sampled by the CCNC – what is the basis for this statement? If there really were losses – can they be quanitifed? What would be the impact on all data measured at the CES observatory? Please add at least some info on intrumental problems and their effects also to this paper.

AR2 4: We absolutely agree that instrumental problems should not be mentioned in half sentences but need to be discussed openly. For that reason we included 10 lines in section 2.2 that describe the problems encountered with the CCNC at the CES observatory. Additionally, we included in the supplementary material a section dedicated to that issue and Fig. S2 shows how the instrumental problem reflects in detail on the closure study. In the manuscript we include therefore a reference: "(see Sec 2.2 and SI Sec 1 for more details)". These passages address the other questions raised in RC2 4. In short, there is no impact on other data at the station because only the CCNC inlet line was affected. The losses of small particles cannot be quantified because there is no size distribution measurement for this inlet line. We state clearly that we show the data uncorrected (p. 10, l. 6) for comparative purposes (p. 23, l. 11) to highlight the importance of quality control to produce high quality data sets.

RC2 5. Disregarding surface tension in the calculation of d\_crit could be problematic. The paper states that haze particles at activation probably have the same surface tension as water, which is not correct (see e.g. Capel et al., 1990, Facchini et al., 2000 and Hitzenberger et al., 2002) and the effect of surface tension on CCN activation, which was indicated e.g. by Charlson et al, 2001, is inadequately discussed.

AR2 5: In the methodology section 2.3.2, we explain that we use the surface tension of water at a temperature of 5 °C and that the surface tension changes with the assumed temperature (see p. 12, l. 6-10). In the closure study results section 3.4, on p. 22, l. 13ff, we discuss that the surface tension is one factor that can play a role in the prediction of CCN number concentrations using the kappa-Köhler theory. And in the introduction on p.7, l.4ff we discussed the effect of the surface tension decrease due to organic surfactants. For that reason, we conducted a sensitivity study based on the JFJ station data and show the results in the supplementary information Sec 2. In the manuscript on p. 24, starting in l. 28 we discuss the results. "Other than the varied parameters shown in Fig. 10a, the value of the surface tension of the solution in the droplet might play a role. Based on JFJ data, using the closure calculations with  $\kappa_{IA+OA-BC}$ , a 30 % decrease (increase) in  $\sigma_{sol}$  would result in a 17 % under-prediction (over-prediction of 25 %, see SI Sec 2) of CCN0.5. This is within the range of change introduced by fixing the particle number concentration or size distribution. However, such a large change in  $\sigma_{sol}$  is not likely as a 30 % decrease can happen if very strong surfactants are present (Petters and Kreidenweis, 2013)."

In summary, we are not disregarding the surface tension, but on purpose choose the simplest approach (i.e. using the surface tension of water) to test the long-term data sets against the kappa-Köhler theory because we do not have more detailed information than the rough aerosol chemical composition (in terms of particulate sulfate, nitrate, chloride, ammonium and bulk organics). As we show, this simplification is justifiable for the type of observatories that we discuss in this study. In addition, we do not state anywhere in the manuscript that "haze particles" have the same surface tension as water. The only time we refer to haze particles is in the context of Arctic haze. Particle properties of Arctic haze are very different from urban haze particles, they are very aged and contain mainly particulate sulfate in form of sulfuric acid or ammonium sulfate (if available). We are not aware of any literature that reports on surface tension depressing organic surfactants in Arctic Haze. The only urban location that we include in this study is Seoul. However, neither the data set of Seoul nor Barrow (Arctic) are included in the kappa-Köhler closure study, because we do not have chemical composition data for these stations.

MC2 5: We include in the introduction, p. 7, l. 5: "In addition, several studies have investigated the effect of organic surfactants that can decrease the surface tensions (e.g., Charlson et al., 2001; Facchini et al., 2000)."

RC2 6. The title of the paper is misleading, as it suggests a far reaching review of what "we" (the scientific community?) have learned about CCN and CCN closure. This is not the case – the paper presents valuable data and valuable data analysis, but it is nevertheless limited to the ACTRIS network plus one station each in Korea and Japan, and two stations in the US. Global coverage is patchy, and as four of the stations are coastal background, three are rural background, two are high alpine, two are remote, and one is urban, the question of how representative these stations are for a global assessment remains open. Please change the title to avoid misunderstandings

AC/MC2 6: We have changed the title to: "Long-term cloud condensation nuclei number concentration, particle number size distribution, and chemical composition measurements at regionally representative observatories"

*RC2 7: Earlier work is referenced somewhat selectively. The paper mentions and references some previously published data sets, too, but implies that most studies were based on short-term intensive field campaigns, or that there are no (short or long-term) data sets for urban areas. This is not*

correct – see the studies by Che et al. (2016) and (2017) and Deng et al. (2013) conducted in more or less polluted regions in China (including also parametrizations of CCN activation), the studies at urban sites published by Burkart et al. (2011) and (2012) for the urban aerosol in Vienna, Austria (the latter study also includes kappa from chemical analyses of particles < 100 nm), and by Leng et al. (2013) in downtown Shanghai; the two year study by Fors et al. (2011) in rural Sweden and for the boreal forest by Sihto et al. (2011). This is just a short list of pertinent studies not referenced in the paper. At least those that include discussion of CCN parameters other than concentrations should be included in the discussion.

AR2 7: It is correct that the field of literature regarding CCN measurements is much larger than what we can represent in this particular study. By no means have we intended to say that there are no studies in urban areas, since several of the co-authors participated in urban studies. We limited references to urban areas because we focus on (but are not limited to) regionally representative sites. We appreciate the reviewer's hints towards other long-term studies and have included more information and references in the introduction.

With regards to the discussion, the reviewer does not point out any specific deficit that we should discuss in this study and that would benefit specifically from a reference to the mentioned studies. Given that both reviewers say that the manuscript is already lengthy, we prefer to keep the discussion as it is.

[revised manuscript text omitted]

Julia Schmale1, Silvia Henning2, Stefano Decesari3, Bas Henzing4, Helmi Keskinen5,6, Mikhail Paramonov5,7, Karine Sellegri8Sellegri7, Jurgita Ovadnevaite9Ovadnevaite8, Mira L. Pöhlker40Pöhlker9, Joel Brito11,8 Brito10,7, Aikaterini Bougiatioti12 Bougiatioti11, Adam Kristensson13 Kristensson12, Nikos Kalivitis12Kalivitis11, Jefferson14Jefferson13 Stavroulas12Stavroulas11, Samara Carbone10Carbone10 Iasonas Anne Park15Park14, Schlag16,17Schlag14,16 Patrick Minsu Yoko 10 Iwamoto18,19Iwamoto17,18, Pasi Aalto5, Mikko Äijälä5, Nicolas Bukowiecki1, Mikael Ehn5, Göran  $\frac{1}{\text{Frank}^{13}\text{Frank}^{12}}, \text{ Roman Fröhlich}^{1}, \text{ Arnoud } \frac{1}{\text{Frumau}^{20}\text{Frumau}^{19}}, \text{ Erik Herrmann}^{1}, \text{ Hartmut Herrmann}^{2}, \text{ Rupert } \frac{1}{\text{Holzinger}^{16}\text{Holzinger}^{15}}, \text{ Gerard } \frac{1}{\text{Kos}^{20}\text{Kos}^{19}}, \text{ Markku Kulmala}^{5}, \text{ Nikolaos}$  $\frac{1}{Mihalopoulos^{12,21}Mihalopoulos^{11,20}}, \text{ Athanasios } \frac{Nenes^{22,21,23}}{Nenes^{21,20,22}}, \text{ Colin } \frac{O'Dowd^9}{O'Dowd^8},$ Tuukka Petäjä5, David Picard8Picard7, Christopher Pöhlker10Pöhlker9, Ulrich Pöschl10Pöschl9, Laurent Poulain2, André Stephan Henry Prévôt1, Erik Swietlicki13Swietlicki12, Meinrat O. Andreae10Andreae9, 15 Paulo Artaxo14Artaxo10, Alfred Wiedensohler2, John Ogren14Ogren13, Atsushi Matsuki18Matsuki17, Seong Soo Yum15Yum14, Frank Stratmann2, Urs Baltensperger1, and Martin Gysel1

1 Laboratory of Atmospheric Chemistry, Paul Scherrer Institute, 5232 Villigen, Switzerland
 2 Leibniz Institute for Tropospheric Research, Permoserstrasse 15, 04318 Leipzig, Germany
 3Institute of Atmospheric Sciences and Climate, National Research Council of Italy, Via Piero Gobetti, 101, 40129 Bologna, Italy

5Department of Physics, University of Helsinki, Gustaf Hällströmin katu 2, 00014, Helsinki, Finland
 6Hyytiälä Forestry Field Station, Hyytiäläntie 124, Korkeakoski, Finland
 7Institute for Atmospheric and Climate Science, ETH Zurich, Universitätsstrasse 16, 8092 Zurich, Switzerland
 8Laboratory 7Laboratory for Meteorological Physics (LaMP), Université Clermont Auvergne, F-63000 Clermont-Ferrand, France

30 9School8School of Physics and CCAPS, National University of Ireland Galway, University Road, Galway, Ireland 10Multiphase9Multiphase Chemistry and Biogeochemistry Departments, Max Planck Institute for Chemistry, Mainz, Germany

11Instituto10Instituto de Física, Universidade de São Paulo, Rua do Matão 1371, CEP 05508-090, São Paulo, SP, Brazil 12Department11Department of Chemistry, University of Crete, Voutes, 71003 Heraklion, Greece

35 13Department12Department of Physics, Lund University, 221 00 Lund, Sweden

15Department14Department of Atmospheric Science, Yonsei University, Seoul, South Korea

16Institute15Institute for Marine and Atmospheric Research, University of Utrecht, Utrecht, The Netherlands

40 47Institute16Institute for Energy and Climate Research (IEK-8): Troposphere, Forschungszentrum Jülich, Jülich, Germany

<sup>4Netherlands Organisation for Applied Scientific Research, Princetonlaan 6, 3584 Utrecht, The Netherlands

<sup>14Earth13Earth System Research Laboratory, National Oceanic and Atmospheric Administration, 325 Broadway, Boulder, CO 80305, USA

18Institute17Institute of Nature and Environmental Technology, Kanazawa University, Kakuma-machi, Kanazawa 920-1192, Japan

49Graduate 18Graduate School of Biosphere Science, Hiroshima University, 1-4-4, Kagamiyama, Higashi-Hiroshima 739-8528, Japan

5 20Energy19Energy Research Center of the Netherlands, Petten, The Netherlands

21National20National Observatory of Athens, P. Penteli 15236, Athens, Greece

22School21School of Chemical & Biomolecular Engineering and School of Atmospheric Sciences, Georgia Institute of Technology, Atlanta, GA, 30332-0340, USA

23Foundation22Foundation for Research and Technology – Hellas, Greece

10

Correspondence to: Julia Schmale (julia.schmale@gmail.com) and Martin Gysel (martin.gysel@psi.ch)

**Abstract.**

Aerosol-cloud interactions (ACI) constitute the single largest uncertainty in anthropogenic radiative forcing. To reduce the uncertainties and gain more confidence in the simulation of ACI, models need to be evaluated against observations, in particular against measurements of cloud condensation nuclei (CCN). Numerous observations of CCN number concentration

- 5 exist, and many closure studies have been performed to predict CCN number concentrations based on particle number size distributions, chemical composition, and the  $\kappa$  Köhler theory. Most of these studies provide details for short time periods or focus on special environmental conditions. These observations, however, cannot address questions of large scale temporal and spatial CCN variability. Here we analyze present a data set - ready to be used for model validation - of long-term observations of CCN number concentrations, particle number size distributions and chemical composition from twelve sites
- 10 on three continents. Eight of these stations are part of the European Aerosols, Clouds, and Trace gases Research InfraStructure (ACTRIS).

We group the observatories into categories according to their official classification: Studied environments include: coastal background (Barrow, Alaska; Mace Head, Ireland; Finokalia, Crete; Noto Peninsula, Japan), rural background (Melpitz, Germany; Cabauw, the Netherlands; Vavihill, Sweden), alpine sites (Puy de Dôme, France; Jungfraujoch, Switzerland),

- 15 remote forests-sites (ATTO, Brazil; SMEAR, Finland) and the an urban environment-surrounding. (Seoul, South Korea). Expectedly, CCN characteristics are highly variable across regionssite categories. However, they also vary within categoriesthem, most strongly in the coastal background group, where CCN number concentrations can vary by up to a factor of 30 within one season. In terms of particle activation behavior, most continental stations exhibit very similar <del>relative</del> activation ratios <del>across the range of 0.1 (relative to 1.0 % supersaturation. particles > 20 nm) across the range of 0.1 to 1.0 %</del>
- 20 supersaturation. At the coastal sites the transition from particles being CCN inactive to becoming CCN active occurs over a wider range of the supersaturation spectrum. At the coastal sites the activation ratios spread more widely across the supersaturation spectrum.

Several stations show strong seasonal cycles of CCN number concentrations and particle number size distributions, e.g., at Barrow (Arctic Haze in spring), at the alpine stations (stronger influence of polluted boundary layer air masses in summer),

- the rain forest (wet and dry season), or Finokalia (forest fire influence in fall). The rural background and urban sites exhibit relatively little variability throughout the year while short-term variability can be high especially at the urban site. The average hygroscopicity parameter, κ, calculated from the chemical composition of submicron particles, was highest at the coastal site of Mace Head (0.6) and the lowest at the rain forest station ATTO (0.2 0.3). We performed closure studies based on the κ-Köhler theory to predict CCN number concentrations. from the particle number size distribution and chemical
- 30 composition measurements. The prediction accuracy for the average concentrations is high. The ratio of predicted to measured CCN concentrations is between 0.87 and  $1.4_{\frac{1}{2}}$  for five different types of  $\kappa$ . The temporal variability is also well represented captured, as reflected by Pearson correlation coefficients > 0.87. We also conducted a series of sensitivity studies for the ratio of predicted versus measured CCN concentration, where we varied the hygroscopicity parameter  $\kappa$ , and made simple assumptions for aerosol particle number concentrations and size distributions. Uncertain particle number

concentrations and their size distributions significantly impair the accuracy in predicting temporal variability and hence of absolute concentrations, while the effect of uncertain κ values is limited to the predicted CCN number concentration.

Information on CCN number concentrations at many locations is important to better characterize ACI and their radiative forcing. But Hong-term comprehensive aerosol particle characterizations are\_-labor intensive and costly. Hence, Wwe recommend operating "migrating-CCNCs" to conduct collocated CCN number concentration and particle number size distribution measurements at individual locations throughout one year at least to derive a seasonally resolved hygroscopicity

parameter. This way, CCN number concentrations can be calculated based on continued particle number size distribution information only and greatgreater spatial coverage of long-term measurements can be achieved.

at priority locations, identified by model evaluation, around the globe where long term particle number size distribution data

- 10 are already available.For observatories where such efforts are out of scope to obtain nevertheless long term information of CCN number concentrations, we recommend conducting collocated CCN number concentration and particle number size distribution measurements at individual locations throughout one year at least to derive a seasonally resolved hygroscopicity parameter. This way, CCN number concentrations can be calculated based on continued particle number size distribution information only. This approach is a good alternative to deriving kappa from time resolved chemical composition
- 15 measurements which are costly and may still not cover the appropriate size range. Additionally, given the variability in observations at sites of the same category, a certain density in spatial coverage of observations is needed, especially along coastlines. We recommend operating "migrating CCNCs" at priority locations, identified by model evaluation, around the globe 
[revised manuscript text omitted]

**10 **2.2 Data treatment and quality assurance**

15

The Ccollection, and harmonization and quality assurance of the data sets presented here is are described in detail in the data descriptor by Schmale et al. (2017). In short, where available, level 2 data for particle number size distributions were downloaded from the EBAS database (http://ebas.nilu.no/). These data are fully quality assured, Data have a time resolution of one hour and represent standard temperature and pressure (STP) conditions. Where not available from EBAS, data were provided by the originators themselves in their preferred data format. This is also true for all CCN and aerosol chemical composition data. The data used here were converted into time series corresponding to the EBAS level 2 format using temperature and pressure information either provided by the data originators or the EBAS database. All time series were avaraged to one hour except for The time resolution of CCN number concentrations at Puv de Dôme (PUX, four hours) and

averaged to one hour except for The time resolution of CCN number concentrations at Puy de Dôme (PUY, four hours) and ATTO (six hours) are four and six hours, respectively, where because the scans over the submicron aerosol size range in monodisperse mode took longer.

The quality assurance of the entire data set is described in Schmale et al. (2017). Briefly, sudden changes and outliers in number concentrations (CCN or chemical composition) or particle sizes were discussed with the data originators and removed if necessary. 
[revised manuscript text omitted]

5

| station | K IA+OA-BC | K IB+OA-BC | K IA+OA+BC | K IB+OA+BC | <del>к = 0.3</del> |
|---------|-----------------------|-----------------------|-----------------------|-----------------------|--------------------|
| ATT     | 0.26                  | 0.21                  | 0.25                  | 0.20                  | <del>0.30</del>    |
| CES     | 0.52                  | 0.50                  | 0.50                  | 0.48                  | <del>0.30</del>    |
| FIK     | 0.48                  | 0.47                  | 0.46                  | 0.45                  | <del>0.30</del>    |
| JFJ     | 0.41                  | 0.31                  | 0.39                  | 0.29                  | <del>0.30</del>    |
| MEL     | 0.43                  | 0.42                  | 0.42                  | 0.42                  | <del>0.30</del>    |
| MHD     | 0.63                  | 0.63                  | 0.61                  | 0.61                  | <del>0.30</del>    |
| SMR     | 0.30                  | 0.29                  | 0.27                  | 0.25                  | <del>0.30</del>    |

10

Table 3: Comparison of geometric and to arithmetic mean values of the ratios of predicted and measured  $CCN_{0.5}$  number concentrations based on calculations with the composition-derived  $\kappa_{IA+OA-BC}$ .

| station | geometric mean | arithmetic mean |
|---------|----------------|-----------------|
| ATT     | 1.06           | 0.94            |
| CES     | 3.10           | 2.31            |
| FIK     | 0.87           | 0.84            |
| JFJ     | 1.09           | 0.93            |
| MEL     | 1.23           | 1.28            |
| MHD     | 1.14           | 1.14            |
| SMR     | 1.32           | 1.19            |

aerosol, organic aerosol mass and black carbon were considered.

**Figures**